

# Practical realization of chiral nonlinearity
# for strong topological protection

Xinxin Guo[1,2], Lucien Jezequel[3], Mathieu Padlewski[1],
Hervé Lissek[1], Pierre Delplace[3] and Romain Fleury[1*]

**1** École Polytechnique Fédérale de Lausanne, Laboratory of Wave Engineering,
CH-1015 Lausanne, Switzerland
**2** ETH Zürich, Combustion, Acoustics & Flow Physics laboratory, 8092 Zürich, Switzerland
**3** École Normale Supérieure de Lyon, CNRS, Laboratoire de physique, F-69342 Lyon, France

⋆ romain.fleury@epfl.ch

## Abstract

Nonlinear topology has been much less inquired compared to its linear counterpart. Existing advances have focused on nonlinearities of limited magnitudes and fairly homogeneous types. As such, the realizations have rarely been concerned with the requirements for nonlinearity. Here we explore nonlinear topological protection by determining nonlinear rules and demonstrate their relevance in real-world experiments. We take advantage of chiral symmetry and identify the condition for its continuation in general nonlinear environments. Applying it to one-dimensional topological lattices, we show possible evolution paths for zero-energy edge states that preserve topologically nontrivial phases regardless of the specifics of the chiral nonlinearities. Based on an acoustic prototype design with non-local nonlinearities, we theoretically, numerically, and experimentally implement the nonlinear topological edge states that persist in all nonlinear degrees and directions without any frequency shift. Our findings unveil a broad family of nonlinearities compatible with topological non-triviality, establishing a solid ground for future drilling in the emergent field of nonlinear topology.



# 1   Introduction

Topological protection has received a surge of interest owing to its strong immunity to parametric perturbations and geometrical defects. It has been investigated on versatile platforms, from quantum mechanics [1] to multifarious classical realms such as electronics [2–5], photonics [6–10] and phononics [11–20]. In contrast to the tremendous attention paid to linear physics and band theory, topological research has less accented on the intersections with nonlinear dynamics [10, 21, 22], despite the ubiquity of nonlinearity in nature. The nonlinear sources exploited for topological purposes include varactor diodes inserted in electrical circuits [4, 5, 23–25], optical materials with intensity-dependent refractive index [10, 26–28], geometry [29, 30] or nonlinear stiffness [31–33] of mechanical structures, and active means that create nonlinearity together with non-Hermiticity [34]. However, the types of nonlinearities are rather homogeneous in previous surveys, with a strong dominance of Kerr-like onsite nonlinearities [5, 6, 10, 23, 26, 27, 31, 33, 35–42], due to their ease in passive realizations and the link to bosonic quantum systems through the well-known Gross-Pitaevskii equation [43]. Exceptions arise mainly from the use of specific lasers [28, 34] or electrical elements [4, 25], whose self-focusing or defocusing behaviors are described by saturable nonlinear gains.

The nonlinear effects, once triggered, have resulted in topologically nontrivial phases that were mostly trivial in the linear regime [10, 22], allowing for many fascinating phenomena such as first- or second-order topological insulators [5, 26, 32, 39], soliton propagation [27–29, 31, 37, 44, 45], and higher harmonic generations [24, 46–48]. Nevertheless, studies reported to date possess their own specific effective range of nonlinearities. Some of them have been restricted to weak nonlinear magnitudes to approach theoretical models and/or to enable theoretical analyses (viable linearization and perturbation methods) [4, 5, 31, 35, 40, 46]. Others, on the other hand, have required nonlinearity strong enough to activate nonlinear states (e.g., solitons) or to localize them clearly (e.g. corner topological states). A few have explored large intervals of nonlinear levels from low to high (before chaos), but with the edge modes/states shifted in frequency [23, 33, 36, 38, 49], ultimately destroying topological phases due to nonlinearity-induced symmetry breaking. Nonlinear topology, discovered within limited contents and extents of nonlinearity, has hardly been discussed from a fundamental nonlinear perspective thus far. That is, taking the stand on topological demands, what nonlinearities are actually needed? Is it feasible in practice to keep topological attributes intact across all nonlinear magnitudes?

To tackle the question, here we unlock limitations to the manipulation of nonlinear topological systems in theory and practice, by satisfying a symmetry that maintains topological non-triviality permanently. Different types of symmetries can enable topological phases of matter [6, 12, 50], including time-reversal symmetry [50], reflection symmetry [51], Parity-Time symmetry [34], chiral symmetry [52, 53] or derived sub-symmetries [54]. Our study utilizes chiral symmetry that is closely related to the emergence of zero-energy topological edge states [55]. We first identify the nonlinear condition for symmetry preservation in general periodic systems. We then introduce eligible nonlinearities in one-dimensional (1D) lattices to alter the linearly produced stationary topological edge states. Their variations are qualitatively predictable assuming chiral nonlinearities with general monotonic dependence on amplitudes. A concrete nonlinear case is finally examined in a theoretical lumped element circuit and in the equivalent active nonlinear acoustic system. We confirm theoretically, numerically, and experimentally that under chiral symmetry, nonlinear edge states can sustain their topologically nontrivial phases while never shifting in frequency.

## 2 Chiral symmetry for general nonlinear periodic systems

In terms of the Hamiltonian $\mathbf{H}$ of the system, and in the presence of arbitrary nonlinearities and non-localities depending on the different degrees of freedom $(a_i, b_j, c_k, \dots)$ contained in the system, chiral symmetry implies that $\Gamma \mathbf{H}(a_i, b_j, c_k, \dots) \Gamma^\dagger = -\mathbf{H}(a_i, b_j, c_k, \dots)$, with $\Gamma$ the chiral operator and $\dagger$ the conjugate transpose [53]. In the chiral base of the degrees of freedom, where $\Gamma = \begin{bmatrix} 1_a & 0 \\ 0 & -1_b \end{bmatrix}$ with $1_a$ and $1_b$ the identity matrices of random sizes, this definition is equivalent to say that $\mathbf{H}(a_i, b_j, c_k, \dots)$ is block off-diagonal, namely

$$\mathbf{H}(a_i, b_j, c_k, \dots) = \begin{bmatrix} 0 & h(a_i, b_j, c_k, \dots) \\ h^\dagger(a_i, b_j, c_k, \dots) & 0 \end{bmatrix}. \tag{1}$$

Notably, there are no specific restrictions on the nonlinearities in $h(a_i, b_j, c_k, \dots)$ in Eq. (1). They can, in principle, take any form, and rely randomly on the system elements, even in a non-local way. The only requirement is that the sites of the same chirality must be uncoupled from each other. Conversely, any nonlinearity that creates couplings among them will inevitably cause symmetry breaking, as is the case with the extensively inquired Kerr-like onsite nonlinearity [5, 6, 10, 23, 26, 27, 33, 35–42].

## 3 Generalized nonlinear topological protection with chiral symmetry

The satisfaction of Eq. (1) allows for chiral symmetry in Hamiltonians of any dimension. For a direct application, we focus on the zero-energy edge states in 1D dimerized lattices, where Eq. (1) is already met by the 2x2 Hamiltonian in the natural base. We start with the linear chiral case, for which, the recurrent relations read: $\eta_L a_n + a_{n+1} = 0$ and $\eta_L b_n + b_{n-1} = 0$, where $a_n$ and $b_n$ are the amplitudes of the two sites of the n-th unit cell, and the N sites $a_n$ ($b_n$) constitute the entire sublattice A (B) of the system. A topologically nontrivial phase is obtained if the hopping ratio $\eta_L$ (ratio between the hopping terms) is smaller than one. The resulting linear topological edge state is displayed in Fig. 1, where the sites $a_n$ carry a decrease in amplitudes along A, with the descent rate fixed by $\eta_L$. The presence of chiral symmetry makes the sites $b_n$ stay stationary, independent of $a_n$.

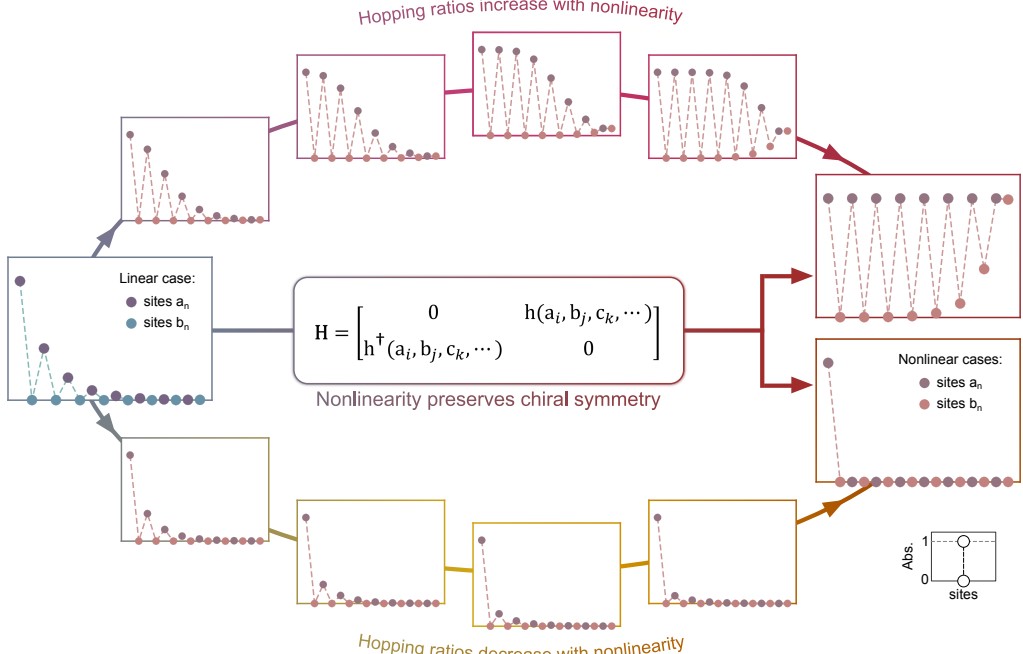

Figure 1: **Qualitative estimations of the evolution laws for zero-energy edge states in 1D dimerized systems with symmetry-preserving nonlinearities.** Profiles of the zero-energy edge state that is initially (linearly) topological and then varied as chiral nonlinearities increase and decrease the hopping ratios on sublattice A, respectively. In each profile, the site amplitudes are normalized such that $a_1 = 1$. The requirement on the Hamiltonian **H** is explained in Eq. (1). The variation trends apply to nonlinearities that lead to monotonic changes in the hopping ratios as the site amplitudes increase, which can also be non-local. When hopping ratios in A increase with nonlinearity (the upper branch), the plateau limit is provided by saturable hopping ratios that do not restrict the associated hopping terms to be of various types such as polynomial (cubic, quadratic, etc.), saturable or even exotics.

When nonlinearities get involved in a way that respects Eq. (1), the system energy relations read:

$$0 = a_{n+1} + \left[ \eta_{\mathrm{L}} + \eta_{\mathrm{NLa}}(a_{n+i}, b_{n+j}) \right] a_n , \qquad 0 = b_{n-1} + \left[ \eta_{\mathrm{L}} + \eta_{\mathrm{NLb}}(a_{n+k}, b_{n+l}) \right] b_n . \quad (2)$$

This suggests that, under chiral symmetry, the participation of nonlinearity results only in modifications in the hopping ratios. They are transformed from the linear invariant $\eta_{\mathrm{L}}$ to the amplitude-dependent nonlinear variables $\eta_{\mathrm{L}} + \eta_{\mathrm{NLa}}(a_{n+i}, b_{n+j})$ applied to $a_n$ and $\eta_{\mathrm{L}} + \eta_{\mathrm{NLb}}(a_{n+k}, b_{n+l})$ applied to $b_n$. $a_{n+i}$ and $b_{n+j}$ ($a_{n+k}$ and $b_{n+l}$) refer to each site that the nonlinearity in $\eta_{\mathrm{NLa}}$ ($\eta_{\mathrm{NLb}}$) depends on. They can be arbitrary in the system, i.e., the integer $i$ or $j$ or $k$ or $l$ can be zero if the dependency occurs within the n-th unit cell, or nonzero if the dependency is on the other interacting unit cells.

Unlike a previous theoretical study discussing one particular form of nonlinearity [53], here we predict edge state variations under generalized chiral nonlinearities and validate them experimentally. We consider the most common relationship between site amplitude and nonlinear effects, namely nonlinearities cause monotonic changes in the hopping ratios with increasing site amplitudes. A broad range of classical nonlinearities satisfy this condition, from polynomial laws (quadratic or cubic, etc.) to saturable effects. In addition, non-linear laws are not restricted to local effects. We first deal with nonlinearities that are positively cor-

related with amplitudes. Based on $a_n > a_{n+1}$ of the linear state, these nonlinearities lead to $|\eta_{\mathrm{NLa}}(a_{n+i}, b_{n+j})| > |\eta_{\mathrm{NLa}}(a_{n+1+i}, b_{n+1+j})|$ in the early nonlinear stage (negligible effects of sites in B, since they carry zero amplitude initially), where the sign of $\eta_{\mathrm{NLa}}$ determines whether nonlinearity increases or decreases the hopping ratios on A. If $\eta_{\mathrm{NLa}} < 0$, we have $\eta_{\mathrm{L}} + \eta_{\mathrm{NLa}}(a_{n+i}, b_{n+j}) < \eta_{\mathrm{L}} + \eta_{\mathrm{NLa}}(a_{n+1+i}, b_{n+1+j}) < \eta_{\mathrm{L}} < 1$, i.e., the hopping ratios are diminished by nonlinearity, with the decrement less and less along A. As nonlinearity is further strengthened, its positive dependence on amplitudes perpetuates the above law. The first hopping ratio remains thus the smallest, always yielding the largest reduction of amplitude from $a_1$ to $a_2$. Following this trend, we reach a limit situation where solely the first site $a_1$ has a nonzero amplitude. The sites $b_n$ in B remain at zero amplitude, owing to chiral symmetry and the fast decay of the nonlinear mode that prevents it from reaching the other end of the system. The total expected edge state variations for nonlinearity decreasing the hopping ratios on A ($\eta_{\mathrm{NLa}} < 0$) are depicted graphically in the lower branch in Fig. 1.

The reasoning applies likewise to the opposite scenario of $\eta_{\mathrm{NLa}} > 0$, where $|\eta_{\mathrm{NLa}}(a_{n+i}, b_{n+j})| > |\eta_{\mathrm{NLa}}(a_{n+1+i}, b_{n+1+j})|$ results in $\eta_{\mathrm{L}} + \eta_{\mathrm{NLa}}(a_{n+i}, b_{n+j}) > \eta_{\mathrm{L}} + \eta_{\mathrm{NLa}}(a_{n+1+i}, b_{n+1+j}) > \eta_{\mathrm{L}}$. Namely the hopping ratios are increased by nonlinearity, with the increment smaller and smaller along A. Remarkably, the first ratio is the largest here, contrary to the previous case of $\eta_{\mathrm{NLa}} < 0$. The enhancement of nonlinearity impels it to first attain 1, at which moment the site $a_2$ acquires the same amplitude as $a_1$. After that, if nonlinearity still can increase the hopping ratio, $a_2$ exceeds $a_1$ in amplitude. The continuation along this direction makes the ascent of $a_2$ incessant and towards an infinite level, inevitably ending with a physical instability of the system. For this reason, if the edge states can be practically realized at all nonlinear magnitudes, the nonlinearity should always keep the first hopping ratio at 1 once $a_2 = a_1$ is reached. The other hopping ratios will follow the same result due to the periodicity of the system. That is, for $\eta_{\mathrm{L}} + \eta_{\mathrm{NLa}}(a_{n+i}, b_{n+j})$ applied to $a_n$, we have $\eta_{\mathrm{L}} + \eta_{\mathrm{NLa}}(a_{n+i}, b_{n+j}) = 1$ once $a_{n+1} = a_n$. Such a saturable nonlinear law in the hopping ratios does not implies that the nonlinear contents in the associated hopping terms should also be the same. Indeed, diverse types of nonlinearities can yield a saturation feature in the ratios, including polynomial (quadratic, cubic, etc.), saturable, or even other exotic ones (e.g., exponential). Ultimately, after successive attainments of $a_{n+1} = a_n$, all sites in A will exhibit the same amplitude, forming a 'plateau' over it.

For actual systems with finite dimensions, the zero-energy mode reaches the other edge of the system when all sites in A are nonlinearly endowed with nonzero amplitudes. In this case, the excited opposite zero mode causes the amplitude of the sites in B to begin to rise, with a lowering from $b_n$ to $b_{n-1}$, i.e., a heightening along the structure. No conclusion can be drawn about the direction of changes in the hopping ratios on B. Their increase or decrease are separate from those on A, as Eq. (1) states. Despite this, it is certain that from an initial value of less than 1, the nonlinearity should drive the hopping ratios up to 1 at most, as we prescribed earlier through the sublattice A. The extreme nonlinear result can hereby be extrapolated: sites $b_n$ conduct an increase in amplitude along B, with merely the first $b_1$ at rest. Our overall estimates for the case of nonlinearity increasing the hopping ratios on A ($\eta_{\mathrm{NLa}} > 0$) are delineated schematically in the upper branch in Fig. 1, where the pattern in B may vary depending on individual circumstances.

Performing the same analysis as above for nonlinearities that are negatively correlated with site amplitudes, one will obtain the same evolution limits as in Fig. 1. Collectively, accounting for a monotonic amplitude dependence of the chiral nonlinearity, it is possible to make the hopping ratios consistently smaller than or at most equal to 1 for both sublattices A and B, hence remain the stationary edge states to be topologically nontrivial at all nonlinear magnitudes. A result similar to part of Fig. 1 was previously observed in a numerical attempt [31], but with a particular nonlinear management and without discussing the underlying symmetry cause. Distinctively, here our starting point is to interrogate chiral symmetry, thus unveiling a

broad class of nonlinearities, though not all, that ensures topological non-triviality, regardless of nonlinear magnitudes.

# 4 A realizable case of nonlinear topological protection with chiral symmetry

To confirm our anticipations in Fig. 1, here we take one example of a concrete finite system and investigate it in a practical configuration. It is represented by the periodic lumped element circuit in Fig. 2a, which consists of $N = 8$ unit cells, each containing linear and nonlinear resonators. The linear resonators $LF_{2k-1}$ and $LF_{2k}$ are identical. They are each made of mass $M_{2k-1}^{(LF)} = M_{2k}^{(LF)}$ and capacitor $C_{2k-1}^{(LF)} = C_{2k}^{(LF)}$, resonating at the frequency $f_{LF}$. For the nonlinear resonators $HF_{2k-1}$ and $HF_{2k}$, their linear components, with mass $M_{2k-1}^{(HF)}$ and $M_{2k}^{(HF)}$, and capacitor $C_{2k-1}^{(HF)}$ and $C_{2k}^{(HF)}$, exhibit resonance at the frequency $f_{HF}$, higher than $f_{LF}$. A larger (linear) resonance bandwidth is assigned to $HF_{2k-1}$ compared to $HF_{2k}$. The generators $V_{2k-1}^{(NL)}$ and $V_{2k}^{(NL)}$ introduce nonlinearity into the resonators $HF_{2k-1}$ and $HF_{2k}$, respectively, with opposite signs in their nonlinear laws:

$$V_{2k-1}^{(NL)} = -G_{NL}\beta_{2k-1}(b_{n-1} + a_n)^2(b_{n-1} - a_n), \qquad V_{2k}^{(NL)} = +G_{NL}\beta_{2k}(a_n + b_n)^2(a_n + b_n), \quad (3)$$

where $G_{NL}$ is a constant parameter with which nonlinearity can be tuned in both magnitudes and directions. And $\beta_{2k-1} = C^{(HF)}/C_{2k-1}^{(HF)}$, $\beta_{2k} = C^{(HF)}/C_{2k}^{(HF)}$ with $C^{(HF)} = (C_{2k-1}^{(HF)} + C_{2k}^{(HF)})/2$.

The physical domain of the system ends with a resonator $HF_{2N+1}$ that follows the features imposed on the other $HF_{2k-1}$, thereby all the $LF_{2k-1}$ ($LF_{2k}$) satisfy the same recurrent dynamic equations (Eq. (A.1) in Appendix A.1), from which one can obtain,

$$\begin{cases} 0 = t_{1b}(b_{n-1}, a_n) b_{n-1} + t_{0b}(b_n, a_n) b_n, \\ 0 = t_{1a}(a_{n+1}, b_n) a_{n+1} + t_{0a}(a_n, b_n) a_n, \end{cases} \quad (4)$$

at two frequencies $f_L$ and $f_H$, where $a_n$ ($b_n$) corresponds to the voltage carried by $LF_{2k-1}$ ($LF_{2k}$). The frequencies $f_L$ and $f_H$ are dominated by the resonances of the resonators $LF_n$ and $HF_n$, respectively, as derived in detail in Appendix A.1 and depicted in Fig. 6 in Appendix B.1. The relations in Eq. (4) are in line with Eq. (2), where the hopping terms take the forms of:

$$\begin{cases} t_{1a}(a_{n+1}, b_n) = C_{2k-1}^{(HF)} + G_{NL}C^{(HF)}\left(a_{n+1}^2 + b_n a_{n+1} - b_n^2\right), \\ t_{0a}(a_n, b_n) = C_{2k}^{(HF)} - G_{NL}C^{(HF)}\left(a_n^2 + b_n a_n - b_n^2\right), \\ t_{1b}(b_{n-1}, a_n) = C_{2k-1}^{(HF)} + G_{NL}C^{(HF)}\left(b_{n-1}^2 + a_n b_{n-1} - a_n^2\right), \\ t_{0b}(b_n, a_n) = C_{2k}^{(HF)} - G_{NL}C^{(HF)}\left(b_n^2 + a_n b_n - a_n^2\right). \end{cases} \quad (5)$$

The hopping ratios $\eta_L + \eta_{NLa}$ and $\eta_L + \eta_{NLb}$ in Eq. (2) can be determined accordingly by $\eta_L + \eta_{NLa} = t_{0a}(a_n, b_n)/t_{1a}(a_{n+1}, b_n)$ and $\eta_L + \eta_{NLb} = t_{0b}(b_n, a_n)/t_{1b}(b_{n-1}, a_n)$.

Theoretically, after deriving Eq. (4), the edge state generations also require closed-closed boundary conditions in the system, i.e., the voltages associated with $b_0$ and $a_e$ in Fig. 2a should be zero. Such a non-driven configuration cannot be easily implemented in practice, especially in the acoustic system we opt for, as excitation is necessary to trigger the edge states experimentally. For this reason, the theoretical exploration here is carried out by directly considering boundary conditions that are feasible in experiments while playing the same role as the ideal $b_0 = a_e = 0$. Specifically, we let $b_0$ and $a_e$ each obey an equivalent non-reflecting boundary condition in planar acoustic wave propagation. On this basis, the (planar wave) excitation is included in $b_0$. The definitions of the overall boundary conditions are detailed in Appendix A.2.

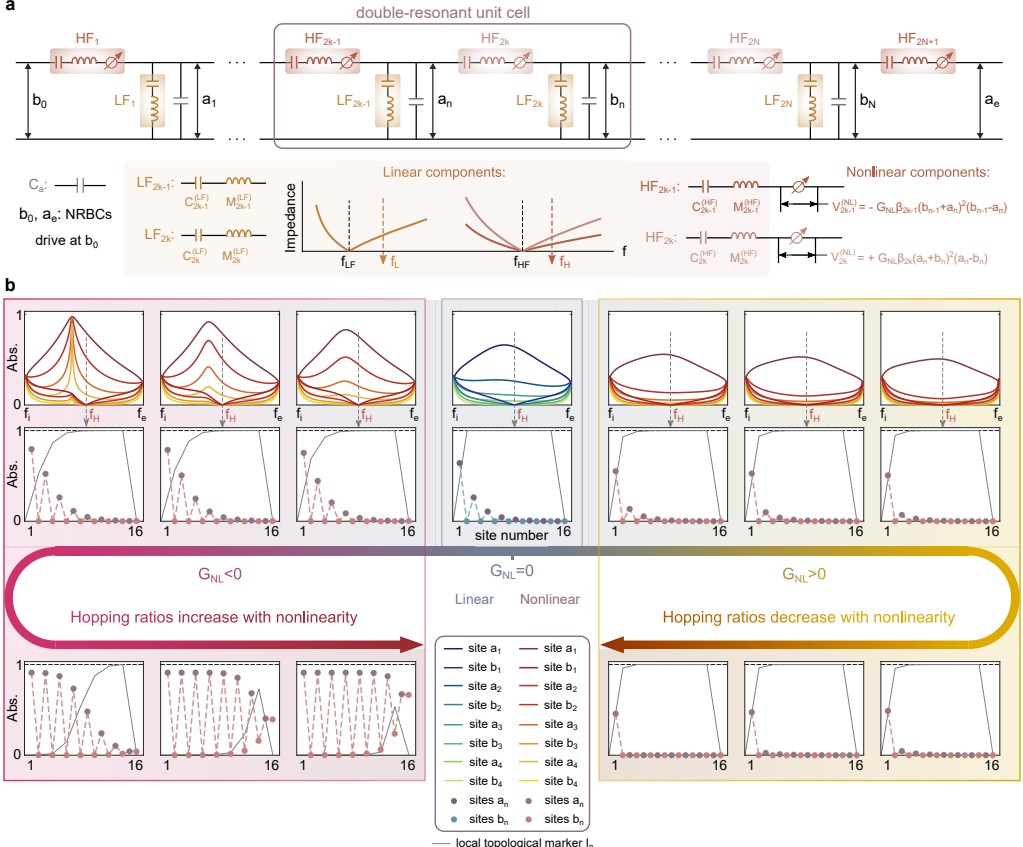

Figure 2: **Evolution of the chiral symmetry protected nonlinear topological edge states: theoretical demonstration in a lumped element circuit with coupled resonators.** (**a**) The considered 1D nonlinear system. It is made of 8 unit cells, each composed of 2 linear resonators $LF_n$ and 2 nonlinear resonators $HF_n$ ($n = 2k - 1$ and $2k$) where nonlinearity is added through the generators $V_n^{(NL)}$. The topological edge state is generated at two different frequencies $f_L$ and $f_H$, which rely on the resonance of the resonators $LF_n$ (at $f_{LF}$) and $HF_n$ (at $f_{HF}$) respectively. The derivations are detailed in Appendix A.1 and B.1 (Fig. 6). (**b**) Nonlinear variations of the linearly generated stationary topological edge state, under the intervention of the nonlinearity given in (**a**). The nonlinear levels and directions are tuned by the constant parameter $G_{NL}$ in the nonlinear law. It increases (decreases) the hopping ratios on sublattice A with $G_{NL} < 0$ ($G_{NL} > 0$). The edge state frequency $f_H$ is identified from the spectra of $a_n$ and $b_n$ ($n = 1, 2, 3, 4$) in the frequency range of $[f_i, f_e]$. All the edge state amplitudes in (**b**) are normalized to the same value. They are obtained with the Harmonic Balance Method (Appendix A.3), and the results for more cases are summarized in Fig. 9. A time domain analysis with the time-integration method is outlined in Appendix B.1 (Fig. 10). In addition to the edge states, the local topological marker [53] $\mathcal{I}_n$ (of the $n$-th unit cell, drawn at the location of each $b_n$) is equally displayed for each case (gray lines), which takes values between 0 (not topological) and 1 (topological, indicated by dashed lines).

They produce the same results as those obtained with $b_0 = a_e = 0$, as proved in Fig. 7 in Appendix B.1, which are therefore undertaken in all the following studies.

With reliable boundary conditions, Eqs. (4) and (5) allow stationary topological edge states to be properly generated in our lumped-element system. In the linear regime where $G_{NL} = 0$, Eq. (5) yields $t_{0a} = t_{0b} = C_{2k}^{(HF)}$ and $t_{1a} = t_{1b} = C_{2k-1}^{(HF)}$, i.e., the hopping terms for $a_n$ ($b_n$) and $a_{n+1}$ ($b_{n-1}$) are directly mapped to the capacitances $C_{2k}^{(HF)}$ in $HF_{2k}$ and $C_{2k-1}^{(HF)}$ in $HF_{2k-1}$. The imposed resonance bandwidth relation between the resonators $HF_{2k}$ and $HF_{2k-1}$ leads to $t_{0a}$ ($t_{0b}$)$<t_{1a}$ ($t_{1b}$), i.e., the hopping ratio $\eta_L = t_{0a}/t_{1a} = t_{0b}/t_{1b}$ smaller than 1, the resulting linear edge state is thus topologically non-trivial. By contrast, in the nonlinear regime where $G_{NL} \neq 0$, the hopping terms are dictated by the nonlinearity engaged. Their amplitude dependence is defined non-local, which is not only on the sites $a_n$ and $b_n$ inside the associated n-th unit cell, but also on the sites $b_{n-1}$ and $a_{n+1}$ in the adjacent ones, as expressed in Eq. (5). Despite this complexity, the chosen nonlinearities make the generated nonlinear edge states rigorously maintain chiral symmetry, since the relations in Eq. (4) (thus Eq. (2)) are consistently satisfied regardless of the nonlinear magnitudes and directions. Interestingly, the expressions of the hopping terms in Eq. (5) indicate that $t_{0a}(a_n, b_n) \neq t_{0b}(b_n, a_n)$ if $G_{NL} \neq 0$, i.e., the coupling between $b_n$ and $a_n$, represented by $t_{0a}$ in the associated Schrodinger equations Eq. (4), is different from the coupling between $a_n$ and $b_n$ represented by $t_{0b}$. Accordingly, the nonlinearity we introduce into the system results in the relevant hopping being non-reciprocal. Such a property has hardly been captured in former research in nonlinear topology, where intentions were mostly placed on approaching reciprocal cases [4].

It should be emphasized that, the results in Eqs. (4) and (5) are obtained without any assumption on any behaviors of the elements in the constituent system. They are derived explicitly at two different frequencies $f_L$ and $f_H$, as mentioned earlier and detailed in Appendix A.1. This implies that we can precisely give rise to two stationary topological edge states within one single one-dimensional lattice. In particular, the two states exhibit also different properties. The one at $f_L$ is significantly more sensitive in response to losses in the dominant resonators $LF_n$, manifesting itself in a severe distortion at a weak loss level, whereas the state at $f_H$ can be nearly immune. Their theoretical comparison is provided in Fig. 8 in Appendix B.1 (the experimental results of the state at $f_L$ are given in Appendix B.3). Our attention here is devoted to the edge state at $f_H$ which can remain intact in the actual experimental conditions. Its evolution is revealed in Fig. 2b. In the initial linear scenario, the hopping ratio is defined at around 0.41 (equal to $C_{2k}^{(HF)}/C_{2k-1}^{(HF)}$). The edge state frequency $f_H$ is recognized from the site spectra in Fig. 2b, at the zero amplitudes of all sites $b_n$. Nonlinearity is then triggered and prescribed using the constant parameter $G_{NL}$ in the nonlinear law. When $G_{NL}$ decreases along negative values, the hopping ratios on sublattice A gradually enlarge. The first ratio keeps receiving the greatest increment. It takes the lead to reach 1, followed by the others in succession. At the very end, the plateau on A is infinitely approached, with solely the last hopping ratio still small. Conversely, in the direction $G_{NL} > 0$, nonlinearity incessantly reduces the hopping ratios on A. The relative descent (with respect to the former site) of $a_2$ remains the largest compared to the other sites. The extreme case of only $a_1$ surviving is also attained. As for the sites $b_n$ in B, their amplitudes rise exclusively after the activation of the opposite zero mode (along $G_{NL} < 0$), presenting the expected increasing order from $b_1$ to $b_8$.

To reveal the topological aspect of the system, we plot for each unit cell $n$, the local topological marker $\mathcal{I}_n$ (Fig. 2b) that generalizes in real space and for finite size systems, the bulk winding number of 1D chiral symmetric insulators [53]. This marker applies to the linearisation of Eq. (2) around a given nonlinear mode and captures the topology of small perturbations around it. It is particularly suitable for systems with inhomogeneous hopping amplitudes, such as ours, where the lattice translation invariance breaks down and the usual bulk winding number, defined in the Brillouin zone, becomes inappropriate. At an interface between a

topological region where $\mathcal{I}_n = 1$ and a topologically trivial region where $\mathcal{I}_n = 0$, the linearised system also develops a zero-energy mode. Due to chiral symmetry, smoothly increasing the amplitude of the nonlinear edge state amounts to adding this linearized zero-mode to the non-linear background zero-mode without changing its frequency [53]. Therefore, high-amplitude nonlinear modes can be obtained by summing up linearized chiral-symmetry-protected topological zero modes captured by $\mathcal{I}_n$. When nonlinear magnitude is increased along $G_{NL} > 0$, the interface between the topological phase where $\mathcal{I}_n = 1$ and the edge where $\mathcal{I}_n$ vanishes becomes sharper. The nonlinear edge mode is thus localized more and more on a single site at the edge. In contrast, when $G_{NL} < 0$, the high amplitude region is associated with a vanishing topological marker, indicating a trivial phase, while low amplitude regions are still topological. The interface zero mode is displaced toward the bulk with increasing nonlinear magnitude along $G_{NL} < 0$. Accordingly, the amplitude rise of the nonlinear mode also shifts toward the bulk, which further displaces the topological transition between $\mathcal{I}_n = 1$ and $\mathcal{I}_n = 0$ in a self-sustaining loop, leading to a plateau shape of the nonlinear edge state in the end.

We confirm with Fig. 2b that in our system, the chiral nonlinearities maintain the topological edge state at its linearly produced frequency $f_H$. Contrarily, if nonlinearity breaks the symmetry, the edge state loses its topological features: its amplitude rises on both sublattices A and B, and its frequency shifts away from $f_H$ (see Fig. 11 in Appendix B.1). The site spectra in Fig. 2b evidence in addition that the amplitude relation of $a_{n+1} < a_n$ is linearly valid over the entire frequency range of $[f_i, f_e]$ displayed therein. It can be nonlinearly transformed up to $a_{n+1} = a_n$ only, as the state at $f_H$ shows. Not surprisingly, if nonlinearity is further enhanced from an already reached $a_{n+1} = a_n$, instability would occur at the related frequency. The left-most spectra in Fig. 2b corresponds to the stability limit of this situation, where the site $a_1$ is caught up by $a_2$ at a frequency different from $f_H$. However, the nonlinear edge state at $f_H$ is perpetually stable, since its variations always satisfy $a_{n+1} \leq a_n$. Collectively, the nonlinear results in Fig. 2b fully demonstrate our inferences for the general context of nonlinearity.

## 5 Experimental validations

After theoretical exploration of the realizable lattice in Fig. 2, an equivalent active nonlinear acoustic system is adopted for experimental validation, as pictured in Fig. 3a. A waveguide is used to connect all the resonant elements. The passive Helmholtz resonators are mounted on its (top) side to play the role of the linear $LF_n$, while electrodynamic loudspeakers are inserted inside and are actively controlled to act as the nonlinear $HF_n$. The control for each loudspeaker involves a feedback loop, which uses as inputs the acoustic pressures measured on both faces of the loudspeaker membrane and returns in real-time an output current to the loudspeaker terminals. The corresponding control law combines (i) a linear part that implements the required impedance for $HF_n$ while compensating for the natural losses in the loudspeaker, and (ii) a nonlinear part that realizes the desired nonlinearities given in Eq. (3), as described in detail in Appendix A.6 and Fig. 5. The achieved active resonators $HF_n$ are fully adjustable and reconfigurable, allowing for replicating the theoretical analysis in Fig. 2b. 8 unit cells are constructed in experiments, each composed of two equally spaced Helmholtz resonators and two equally spaced active loudspeakers. After the last cell, one additional loudspeaker is inserted and controlled to achieve the last nonlinear resonator $HF_{2N+1}$ in Fig. 2a, which is necessary for satisfying Eqs. (4) and (5), as explained in the previous section and Appendix A.1. Therefore, there are in total 17 active loudspeakers and 16 passive Helmholtz resonators in the physical domain of the experimental system. The portions of each volume enclosed by adjacent loudspeakers are denoted as $V_a$ in Fig. 3a. Their behaviors on the sub-wavelength scales are similar to that of capacitors, acting as $C_a$ in the theoretical model in

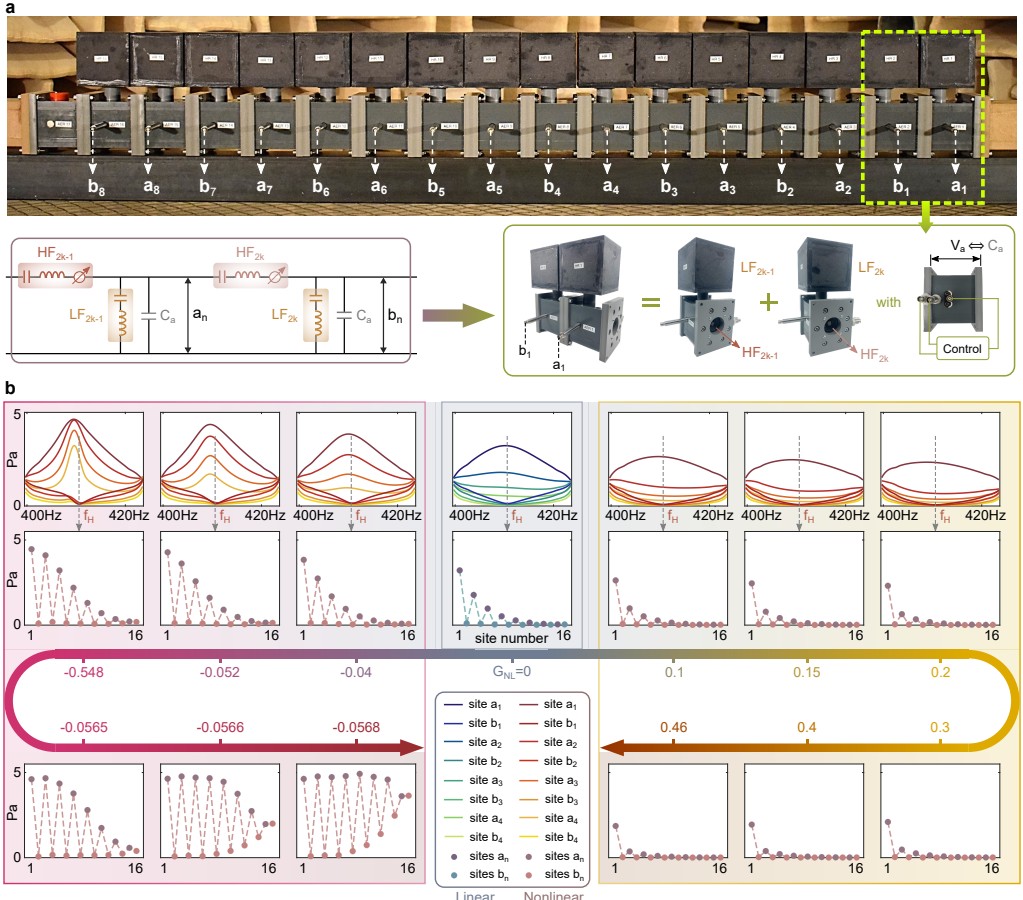

Figure 3: **Evolution of the chiral symmetry protected nonlinear topological edge state: experimental validation in an active nonlinear acoustic system.** (**a**) The actual system that realizes the theoretical lattice in Fig. 2a. The unit cell consists of two passive linear Helmholtz resonators (acting as $LF_n$) and two active nonlinear loudspeakers (acting as $HF_n$). There are two anechoic ends installed at the ends of the system, together with an excitation source at one end. The entire domain of the system is displayed in Fig. 4 in Appendix A.5. The $a_n$ and $b_n$ correspond to the acoustic pressures applied to the Helmholtz resonators $LF_{2k-1}$ and $LF_{2k}$, respectively. (**b**) Nonlinear topological edge states, measured as nonlinearity is progressively altered using the constant control parameter $G_{NL}$. The hopping ratios on sublattice A are increased (decreased) along $G_{NL} < 0$ ($G_{NL} > 0$). The edge state frequency $f_H$ is identified from the spectra of $a_i$ and $b_i$ ($i = 1, 2, 3, 4$). Experimental results of more nonlinear cases are given in Fig. 15 in Appendix B.3.

Fig. 2a. Collectively, the proposed acoustic system is equivalent to the theoretical lattice in Fig. 2a, as sketched in Fig. 3a (see derivations in Appendix A.6). A non-reflecting anechoic termination is installed at each end of the physical domain, with the excitation source fixed at one end. The desired boundary conditions (Appendix A.2) for $b_0$ and $a_e$ in Fig. 2a are accordingly put into practice. The entire system including all the components, is described more in detail in Appendix A.5 and illustrated in Fig. 4 therein.

We focus on the topological edge state at $f_H$ in the main text, as stated in the theoretical studies. It is successfully implemented first in the linear case, as illustrated in Fig. 3b (detailed linear results in Fig. 14 in Appendix B.3). A hopping ratio of around 0.54 is obtained, not very far from the theoretical one of 0.41 (Fig. 2b). The discrepancy stems from the approximation of

each space $V_a$ as a lumped element (the capacitor $C_a$ in the theoretical model), see explanations in Appendix A.6 and demonstrations with simulation results in Appendix B.2. Based on the linear results, nonlinearity is added to the system and tailored by the constant parameter $G_{NL}$, as theoretically set in Fig. 2b. When nonlinear magnitude is reinforced along $G_{NL} < 0$, the hopping ratios on A increase. The sites $a_n$ sequentially attain the same level, enabling the theoretical plateau limit at the greatest extent of nonlinearity. In the meantime of the ascent on sublattice A, the sites in B first remain at rest and then rise in amplitude from the last one $b_8$, which comply also with the theoretical projections.

For nonlinearity decreasing the hopping ratios with $G_{NL} > 0$, the shape of the edge state is centralized more and more on the structure (left) end, with all sites in B staying stationary. The nonlinear variation along this direction proceeds until the first hopping ratio (the smallest one) on A falls to about 0.2, with respect to the linear one of 0.54. The limit of only $a_1$ being dynamic cannot be observed, as a physical instability arises experimentally. This is caused by the time delay unavoidable in feedback control on the loudspeakers, which injects energy into each space $V_a$ enclosed by adjacent loudspeakers. When going beyond the limit case shown in Fig. 3b, the energy accumulations due to such a control delay cannot be fully compensated for, thereby leading to physical instability undoubtedly. A time-domain analysis is performed in Appendix B.2 where the control time delay is taken into account in real-time in the simulations. The relevant numerical results are shown in Fig. 12, which confirms the current experimental observations. Nevertheless, all expected laws of variations are exhaustively justified by experiments. The realized nonlinear edge state preserves its topologically nontrivial phases, with frequency unchanged at $f_H$, since chiral symmetry is here rigorously obeyed by nonlinearity. In the opposite situation where nonlinearity breaks chiral symmetry, the edge state will consequently be distorted in shape and shifted in frequency, as evidenced by theoretical, numerical, and experimental demonstrations in Figs. 11, 13, and 16 in Appendix B, respectively. In addition to inappropriate nonlinearities, non-negligible losses can also break chiral symmetry, as in the case of the second edge state at $f_L$. Its experimental results are summarized in Appendix B.3 (Figs. 17 and 18), in which the persistent dominance of loss effect dramatically disrupt topological properties, as in the theoretical prediction (Fig. 8). Contrary to it, the edge state at $f_H$ is barely affected by actual losses in the system, thanks to the active control by which the dependent loudspeakers are corrected to be virtually loss-free.

## 6 Conclusion

In this study, we explored the nonlinear possibilities for the persistence of topological nontriviality. We targeted the symmetry-protected topological class and put the emphasis on chiral symmetry. The condition to secure symmetry was first formulated for general nonlinear periodic systems. It was then applied to one-dimensional lattices in which zero-energy topological edge states were modified by chiral nonlinearities. The trajectories of their nonlinear evolution were predicted based on a monotonic amplitude dependence of the nonlinearities. The results show that chiral symmetry can consistently maintain the edge states in a topologically nontrivial phase in the nonlinear regime, regardless of the explicit forms and magnitudes of the nonlinearities, whether local or non-local. The derived nonlinear topological edge states were put into practice through the consideration of a concrete finite system, with theoretical representation in a lumped element circuit, and with numerical (Appendix A.4 and B.2) and experimental implementations in an equivalent active nonlinear acoustic system. Our investigations reveal a broad class of chiral nonlinearities that keep the topological attributes intact and the edge state frequency unshifted across all nonlinear magnitudes, opening up new avenues of thought for the continued study of nonlinear topology.

# Acknowledgments

**Author contributions**   R.F. and P.D. initiated and supervised the project. H.L. supervised the experimental work. X.G. established the theoretical modeling, performed the numerical simulations, designed the prototype, and carried out the measurements and data analysis. X.G. and M.P. set up the experiment. L.J. and P.D. developed the theory. X.G. and H.L. raised part of the funding that supported the experiment. All authors contributed to the writing of the manuscript and thoroughly discussed the results.

**Funding information**   X.G., M.P, R.F., and H.L. acknowledge the Swiss National Science Foundation (SNSF) under grant No. 200020_200498. L.J. is funded by a PhD grant allocation Contrat doctoral Normalien.

# A   Methods

## A.1   Achievement of a topological system with chiral symmetry

The dynamics of the lumped-element circuit in Fig. 2a is described in the time domain by

$$
\begin{cases}
\Delta_{2k-1}^{(HF)} q_{2k-1} + C_{2k-1}^{(HF)} V_{2k-1}^{(NL)} = C_{2k-1}^{(HF)} (b_{n-1} - a_n) \,, \\
\Delta_{2k}^{(HF)} q_{2k} + C_{2k}^{(HF)} V_{2k}^{(NL)} = C_{2k}^{(HF)} (a_n - b_n) \,, \\
\Delta_{2k-1}^{(LF)} \left( q_{2k-1} - q_{2k} - q_{2k-1}^{(a)} \right) = C_{2k-1}^{(LF)} a_n \,, \\
\Delta_{2k}^{(LF)} \left( q_{2k} - q_{2k+1} - q_{2k}^{(a)} \right) = C_{2k}^{(LF)} b_n \,.
\end{cases}
\tag{A.1}
$$

Taking into account the expressions of the nonlinear voltage generators $V_{2k-1}^{(NL)}$ and $V_{2k}^{(NL)}$ given in Eq. (3), Eq. (A.1) yields

$$
\begin{cases}
\Delta_t^{(HF)} q_{2k-1} = C_{2k-1}^{(HF)} (b_{n-1} - a_n) + G_{NL} C^{(HF)} (b_{n-1} + a_n)^2 (b_{n-1} - a_n) \,, \\
\Delta_t^{(HF)} q_{2k} = C_{2k}^{(HF)} (a_n - b_n) - G_{NL} C^{(HF)} (a_n + b_n)^2 (a_n - b_n) \,, \\
\Delta_t^{(LF)} \left( q_{2k-1} - q_{2k} - q_{2k-1}^{(a)} \right) = C_{2k-1}^{(LF)} a_n \,, \\
\Delta_t^{(LF)} \left( q_{2k} - q_{2k+1} - q_{2k}^{(a)} \right) = C_{2k}^{(LF)} b_n \,.
\end{cases}
\tag{A.2}
$$

The time-domain variables in Eq. (A.2) include:
(I) The $q_{2k-1}$, $q_{2k}$ and $q_n^{(a)}$ ($n = 2k-1$ and $n = 2k$), which designate the charges of the resonators $HF_{2k-1}$, $HF_{2k}$ and the capacitor $C_a$ in Fig. 2a, respectively.

(II) The voltages applied to $LF_{2k-1}$ and $LF_{2k}$ that are equivalent to $a_n$ and $b_n$ in the topological dimerized lattice, as delineated in Fig. 2a.

(III), The time-domain differential operators $\Delta_t^{(HF)}$ and $\Delta_t^{(LF)}$, which read

$$
\begin{cases}
\Delta_t^{(HF)} = \left[ M_{2k-1}^{(HF)} C_{2k-1}^{(HF)} \dfrac{d^2}{dt^2} + 1 \right] = \left[ M_{2k}^{(HF)} C_{2k}^{(HF)} \dfrac{d^2}{dt^2} + 1 \right] \,, \\
\Delta_t^{(LF)} = \left[ M_{2k-1}^{(LF)} C_{2k-1}^{(LF)} \dfrac{d^2}{dt^2} + 1 \right] = \left[ M_{2k}^{(LF)} C_{2k}^{(LF)} \dfrac{d^2}{dt^2} + 1 \right] \,,
\end{cases}
\tag{A.3}
$$

since all the resonators $LF_{2k-1}$ and $LF_{2k}$ resonate at the same frequency $f_{LF}$, while all the $HF_{2k-1}$ and $HF_{2k}$ resonate at $f_{HF}$.

Substituting Eq. (A.3) into Eq. (A.2) and eliminating all terms containing charges, the equations on voltages can be obtained as follows:

$$
\begin{cases}
\Delta_t a_n = \Delta_t^{(LF)} \left[ C_1^{(HF)} b_{n-1} + C_2^{(HF)} b_n - C_1^{(HF)} V_{2k-1}^{(NL)} + C_2^{(HF)} V_{2k}^{(NL)} \right], \\
\Delta_t b_n = \Delta_t^{(LF)} \left[ C_1^{(HF)} a_{n+1} + C_2^{(HF)} a_n + C_1^{(HF)} V_{2k+1}^{(NL)} - C_2^{(HF)} V_{2k}^{(NL)} \right],
\end{cases}
\quad \text{(A.4)}
$$

where $C_1^{(HF)} = C_{2k-1}^{(HF)}$, $C_2^{(HF)} = C_{2k}^{(HF)}$, and where the fourth-order differential operator $\Delta_t$ takes the form of $\Delta_t^{(HF)} \Delta_t^{(LF)} C_a + \Delta_t^{(HF)} C^{(LF)} + 2C^{(HF)} \Delta_t^{(LF)}$.

It is worth noticing that, to derive Eq. (A.4) also for the last one $b_N$, there should be an additional resonator $HF_{2N+1}$ which makes the associated charge $q_{2N+1}$ also satisfy the first equation in Eq. (A.2). Hence, the physical domain of the system should start with a $HF_1$ and end with a $HF_{2N+1}$. Under this circumstance, if $\Delta_t$ in Eq. (A.4) can be zero, it leads to Eq. (4) in Section 4, with the hopping terms expressed in Eq. (5). This suggests that the stationary topological edge state can be generated from a solution for $\Delta_t = 0$. Focusing on the fundamental frequency $\omega$ at where we have $\frac{d}{dt} = i\omega$ (with i the complex unit), $\Delta_t = 0$ is transformed to the frequency domain as:

$$
\omega^4 - \left[ \left( 1 + 2\frac{C^{(HF)}}{C_a} \right) \omega_{HF}^2 + \left( 1 + \frac{C^{(LF)}}{C_a} \right) \omega_{LF}^2 \right] \omega^2 + \omega_{HF}^2 \omega_{LF}^2 \left[ 1 + 2\frac{C^{(HF)}}{C_a} + \frac{C^{(LF)}}{C_a} \right] = 0, \quad \text{(A.5)}
$$

where $\omega_{HF} = 2\pi f_{HF}$ and $\omega_{LF} = 2\pi f_{LF}$ are the resonance frequencies of the resonators $HF_n$ and $LF_n$, respectively. And $C^{(LF)} = C_{2k-1}^{(LF)} = C_{2k}^{(LF)}$, $C^{(HF)} = (C_{2k-1}^{(HF)} + C_{2k}^{(HF)})/2$.

Interestingly, the determinant of the equation in Eq. (A.5) is strictly positive, i.e., $\Delta_t = 0$ presents two solutions. This is why our system allows for topological edge states at two different frequencies (at $f_L$ and $f_H$), see Fig. 6 in Appendix B.1 for physical explanations. In all edge state profiles shown in this study, $a_n$ and $b_n$ correspond to the absolute amplitudes extracted at the fundamental frequency. Regarding the higher harmonic generations, we confirm theoretically, numerically, and experimentally that they are consistently lower than 1% in achieving the edge state at $f_H$. Thus we can assume no energy conversion from the fundamental component to the higher harmonics in our system, in which case, the derivation of Eq. (A.5) using $\frac{d}{dt} = i\omega$ holds directly.

## A.2 Boundary conditions

In our search for applicable boundary conditions, we eventually found that the typical Non-Reflecting Boundary Conditions (NRBCs) in planar acoustic wave propagation/excitation can replace the ideal one of $b_0 = a_e = 0$. Based on an electro-acoustic analog where electrical (voltage, current) is equivalent to acoustic (pressure, flow), NRBCs are translated into $a_e = \gamma_a i_{2N+1}$ ($b_0 = \gamma_a i_1$) for the right (left) end of the system in the non-driven case, in which $i_{2N+1}$ ($i_1$) represents the electrical current circulating in $HF_{2N+1}$ ($HF_1$), and $\gamma_a = Z_c/S$ with $Z_c$ the specific acoustic impedance of the air and $S$ the surface area of the propagation medium. When excitation is taken into account in addition, the corresponding end is subjected to a total acoustic pressure, which includes the incoming source $p_{inc}$ and the wave reflected by the physical domain $p_{ref} = p_{inc} - \gamma_a i_1$ (NRBC forces no reflection in the direction of incidence). Collectively, for the case of excitation at $b_0$, the boundary conditions are defined as $b_0 = p_{inc} + p_{ref} = 2p_{inc} - \gamma_a i_1$ and $a_e = \gamma_a i_{2N+1}$. We prove with Fig. 7 in Appendix B.1 that these conditions are equivalent to the theoretically required $b_0 = a_e = 0$, in terms of the edge state generations. They are thus undertaken for all the studies of the concrete theoretical model and the equivalent experimental acoustic system.

### A.3 Methods for theoretical solvings

To solve the problem associated with the circuit in Fig. 2a, we consider the original dynamic equations in Eq. A.1 where all the variables are time-dependent. Two standard methods are exploited for solving these nonlinear differential equations, namely the harmonic balance method [56–58] and the time-integration method [59]. They are capable of handling strong levels of nonlinearities, in contrast to the perturbation method and the method of multiple scales that are valid only at weak nonlinearities.

The Harmonic Balance Method (HBM) refers to a semi-analytical method [56–58] which determines the steady-state solutions of the nonlinear problem. The first 27 harmonics of each variable are taken into account when solving Eq. A.1. The outcomes show that the higher harmonic generations are lower than 1% in our system. The $a_n$ and $b_n$ in Fig. 2b correspond to the absolute amplitudes of their fundamental harmonic components (at the edge state frequency $f_H$). The detailed results (more nonlinear cases than in Fig. 2b) are summarised in Fig. 9 in Appendix B.1.

The time integration method, with the fourth-order Runge-Kutta (RK4), is utilized to solve the problem directly in the time domain, which accounts for the transient responses. The relevant results are given in Fig. 10 in Appendix B.1.

### A.4 Time-domain simulation of the experiments.

To better guide and analyze the experiments, we performed time-domain simulations for the active nonlinear acoustic system built in practice (Fig. 3a). The approach involves a Finite Difference Time Domain (FDTD) method by discretization of the 1D wave equations. Practical details are accounted for in the simulations, that is (i) we consider the wave propagation inside each space $V_a$ between two adjacent loudspeakers (with FDTD), (ii) we add losses in all the resonant elements and the transmission medium according to the experimentally estimated values, (iii) we simulate the actual active control on each loudspeaker where a time delay exists, which is experimentally determined to be $100\,\mu s$. The details of the control principle and definition are described in the following section A.6. Regarding the numerical settings, we randomly take the experimental values of one loudspeaker to define all the others. The simulation outcomes are summed up in Fig. 12 in Appendix B.2. They are essentially identical to the experimental ones.

### A.5 Characterisations of the experimental setup

The overall experimental system is pictured in Fig. 4a, where the non-reflecting boundary conditions are achieved with anechoic terminations at both ends of the system. They are qualified by absorption coefficients higher than 0.998 from 140 Hz (less than 5% of reflection), as shown in Fig. 4b. The waveguide refers to a PVC duct with a cross-sectional area of $6\,cm \times 6\,cm$, which ensures planar wave propagation until 2.86 kHz. The manufactured Helmholtz resonators (labeled with $HR_n$ in Fig. 3a) reach a transmission coefficient of around 0.08 at their resonance frequencies in the range of $[110.5\,Hz, 111.5\,Hz]$, corresponding to an acoustic resistance of $0.005 Z_c$ with $Z_c$ the specific acoustic impedance of the air. The electrodynamic loudspeakers are all the same commercially available Visaton FRWS 5 SC model, while they possess different resonance frequencies (within $[345\,Hz, 375\,Hz]$) and bandwidths, which we calibrated beforehand.



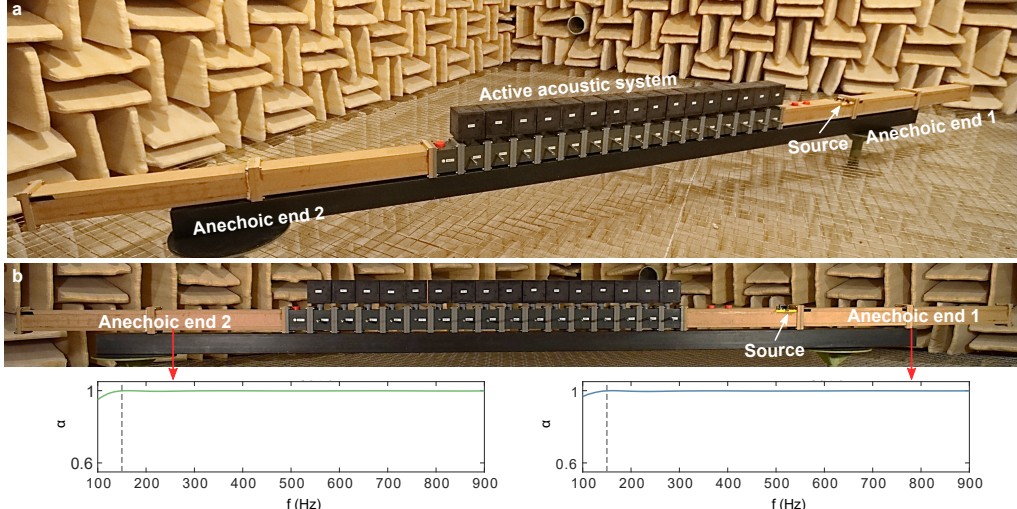

Figure 4: **The overall experimental setup.** Besides the physical domain displayed in Fig. 3a, the overall system includes also two anechoic terminations at the ends, for which the absorption coefficients $\alpha$ are measured and shown in (**b**). The driven source refers to a loudspeaker that is mounted on the top side of the duct next to the anechoic end numbered 1 in (**a**), it plays the role of $b_0$ in Fig. 2a together with the end 1. The boundary $a_e$ is realized by the end number 2.

## A.6 Active control on the electrodynamic loudspeakers.

The loudspeaker membrane behaves as a mass-spring-damper system in the linear regime (weak input levels). The motion equation for its displacement $\xi$ read

$$M_{ms}\frac{\partial^2}{\partial t^2}\xi(t) + R_{ms}\frac{\partial}{\partial t}\xi(t) + \frac{1}{C_{ms}}\xi(t) = p_{tot}(t)S_d - Bli(t). \tag{A.6}$$

In the passive open-circuit case, the membrane is subject to the total acoustic pressure $p_{tot}$ over its effective surface area $S_d$, and the mechanical forces which rely on the mechanical mass $M_{ms}$, resistance $R_{ms}$, and compliance $C_{ms}$. Its dynamics are characterized by a specific acoustic impedance $Z_s$ (ratio between acoustic pressure and velocity) in the frequency domain: $Z_s(j\omega) = \frac{1}{S_d}\left(j\omega M_{ms} + R_{ms} + \frac{1}{j\omega C_{ms}}\right)$.

The active control on each loudspeaker is implemented by specifying the current $i(t)$, which creates an electromagnetic force through the moving coil with a force factor of $Bl$. The control approach is depicted in detail in Fig. 5, where the control law is digitally defined with a Speedgoat real-time target machine manipulated in the Simulink environment of MATLAB. It produces the current $i(t)$ in the form of,

$$i(t) = \mathcal{F}^{-1}\left(\Phi(j\omega) \cdot P_{tot}(j\omega)\right) + \mathcal{F}^{-1}\left(\frac{S_d}{Bl} - \Phi(j\omega)\right) * \left((-1)^n \frac{C^{(exp)}}{C_n^{(exp)}} G_{NL}p_{tot}(t)\left(p_f(t) + p_b(t)\right)^2\right), \tag{A.7}$$

where $p_f$ and $p_b$ are the acoustic pressures measured at the front and rear faces of the loudspeaker membrane, which are the two inputs for the control. $\mathcal{F}^{-1}$ and the symbol $*$ designate the inverse of the Fourier Transform and the time convolution, respectively. The total acoustic pressure $p_{tot}$ reads $p_{tot} = p_f(t) - p_b(t)$, with $P_{tot} = \mathcal{F}(p_{tot})$ its Fourier transform. $C_n^{(exp)}$ refers to the acoustic compliance achieved for the $n$-th loudspeaker which differs between $n = 2k-1$ and $n = 2k$, and $C^{(exp)}$ is the average of two successive ones. $C_{2k-1}^{(exp)}$ and $C_{2k}^{(exp)}$ are equivalent to the electrical capacitors $C_{2k-1}^{(HF)}$ and $C_{2k}^{(HF)}$ in Eq. (A.2), respectively.



Figure 5: **Active control on the loudspeakers.** The linear part of the control is used for altering the impedance $Z_n$ of each loudspeaker to make them resonate at the same frequency while achieving different resonance bandwidths between odd and even ones. The nonlinear part of the control is for producing the nonlinear generators $V_n^{(NL)}$ needed in the theoretical lattice in Fig. 2a. ADC (DAC) denotes the Analog-Digital (Digital-Analog) Converter. A control time delay exists mainly due to the AD and DA conversions and is thus unavoidable for the control law definition. We compensate for this delay effect by carefully defining the control laws, see implementation details in Appendix A.6.

In Eq. (A.7), the linear part of control is represented by a linear transfer function $\Phi(j\omega)$, whereas the nonlinear part is determined by the parameter $G_{NL}$. For the linear part, $\Phi(j\omega)$ is used to tailor the impedance properties of the loudspeaker,

$$\Phi = \frac{S_d}{Bl} \cdot \beta \frac{Z_{st}(j\omega) - Z_s(j\omega)}{Z_{st}(j\omega)} . \tag{A.8}$$

It targets a specific acoustic impedance $Z_{st}^{(F)}$ with two degrees of freedom,

$$Z_{st}^{(F)} = \frac{Z_{st} Z_s}{(1-\beta) Z_{st} + \beta Z_s} = \left[ \frac{1-\beta}{Z_{st}} + \frac{\beta}{Z_s} \right]^{-1} , \tag{A.9}$$

in which the control-designed impedance $Z_{st}$ corresponds to a one-degree-of-freedom resonator. It is made in parallel with the passive one $Z_s$, while their weights are adjusted by the constant parameter $\beta$.

For the control execution, there exists a time delay $\tau$ from control inputs to outputs, which is unavoidable in reality. It is taken into account in simulating the practical case by transforming i(t) into i(t $-\tau$) for Eq. (A.6), and is experimentally determined at $100\,\mu s$. Since the loudspeakers are naturally different, the control time delay affects them differently, yielding discrepancies in control results. Nevertheless, the addition of the parameter $\beta$ in the linear control law allows such an issue to be compensated for in experiments, by balancing between $Z_{st}$ and $Z_s$. Fig. 14 in Appendix B.1 shows the results for linearly generated topological edge state at $f_H$. As for the nonlinear part of the control law in Eq. (A.7), when the sub-wavelength cavity $V_a$ between adjacent loudspeakers exhibits predominantly capacitor characteristics (the assumption under consideration, see Fig. 3a), we have $p_f = b_{n-1}$ and $p_b = a_n$ for loudspeakers with even indexes, and $p_f = a_n$ and $p_b = b_n$ for those with odd indexes. In this case, the nonlinear laws perfectly achieve the generators $V_{2k-1}^{NL}$ and $V_{2k}^{NL}$ (Eq. (3)) required in the theoretical lattice in Fig. 2a.

Performing the above hybrid (linear and nonlinear) control on each loudspeaker, they all become Active Electroacoustic Resonators [60–63] (labeled with $AER_n$ in Fig. 3a), presenting the desired properties for realizing $HF_n$. A low level of less than $1\,Pa$ is maintained for system excitation. It ensures the linear behaviors of the loudspeakers in the passive (control off) regime. Thus, nonlinearity is generated and tuned in an exact way, i.e., through the active control only (using the constant parameter $G_{NL}$ in the control law). The time responses of $a_n$

and $b_n$ are measured by the microphones below Helmholtz resonators, as indicated in Fig. 3a. The edge states shown in Fig. 3b refer to their components at the fundamental frequency (edge state frequency $f_H$). We confirm with measurements that the higher harmonic generations are consistently less than 1% in our acoustic system, which is in line with the theoretical model. The detailed experimental results for the nonlinear topological edge state at $f_H$ are provided in Fig. 15 in Appendix B.3. The cases where nonlinearities break chiral symmetry are investigated in Appendix B. Figs. 11, 13 and 16 show the corresponding theoretical, numerical, and experimental results, respectively. In addition to the state at $f_H$ that we have focused on in the main text, the results for the other edge state at $f_L$ are summarized in Appendix B.3. In Fig. 8, we theoretically demonstrate that the edge state at $f_L$ is extremely sensitive to the losses in the dominant passive resonators $LF_n$. In Figs. 17 and 18, we prove with experimental observations that the actual losses, even already very weak, still cause the state $f_L$ to be severely distorted, see Appendix B.3 for more explanations.

# B  Supplementary results

Here in the Appendix B.1, B.2 and B.3, we show supplementary results for the theoretical study of the lumped-element model in section 4, the time-domain simulations of the actual acoustic system, and the experimental realizations in section 5, respectively.

## B.1  Theoretical results

This section includes:

Fig. 6: Principle and physical explanations for generating dual-band topological edge states in our lumped-element system illustrated in Fig. 2a. Indeed, the impedance of the resonators $HF_n$, having the form of $i\omega M_n^{(HF)} + 1/(i\omega C_n^{(HF)})$, is dominated by the terms $1/(i\omega C_n^{(HF)})$ at frequency much lower than its resonance frequency $f_{HF}$ (small value of $\omega$), which is taken place when close to $f_{LF}$, since we impose $f_{LF} < f_{HF}$ in our system. In contrast, the impedance of the resonators $LF_n$, reading $i\omega M_n^{(LF)} + 1/(i\omega C_n^{(LF)})$, is dominated by the terms $i\omega M_n^{(LF)}$ at frequency much higher than its resonance frequency $f_{LF}$, which occurs at the vicinity of $f_{HF}$. For these reasons, only one type of resonances, $LF_n$ or $HF_n$, can actually act, depending on whether the frequency is close to $f_{LF}$ or $f_{HF}$ (the other resonance behaves as either a capacitor or a mass, as aforementioned). Accordingly, our lumped-element model is equivalent to a system made of single-resonant unit cells at two different frequencies, as depicted in Fig. 6. If $C_{2k}^{(HF)} < C_{2k-1}^{(HF)}$ is always met, then the two approximate cases correspond each to a classic topological lattice. This suggests that one can simplify the system by removing for instance the resonators $LF_n$, which would also produce the recurrent relations that we derived for the current system in Eq. (4). However, when combining the two types of resonances, it is possible to achieve, within one single lattice, two topological edge states presenting completely different features, which motivated the development of our current system. The one we studied in the main text shows robustness in response to all potential losses in the system, whereas the other one at $f_L$ is exceedingly sensitive to losses in the dominant resonators LF, as witnessed by Fig. 8 in Appendix B.1. The experimental results for the state at $f_L$ are outlined in Figs. 17 and 18.

Fig. 7: Proof of the equivalence between the theoretically ideal non-driven boundary conditions $b_0 = a_e = 0$ (Fig. 7b) and the non-reflecting driven ones (Fig. 7c) that are more realizable for experimental realizations. The latter is derived and explained in detail in Section A.2.

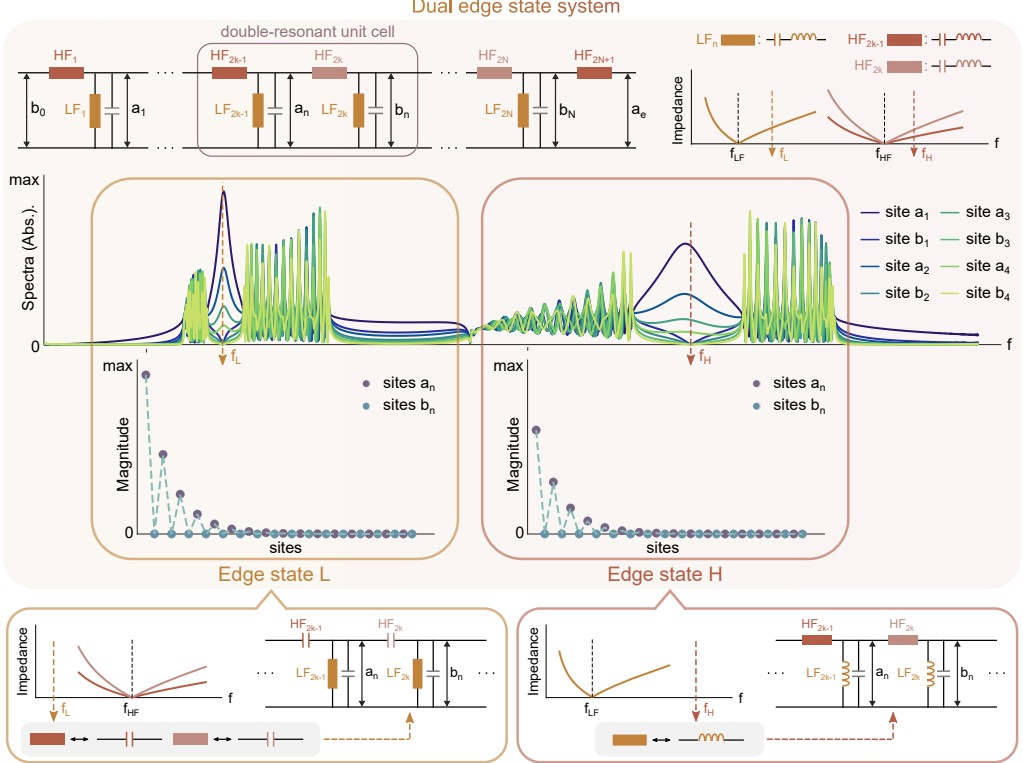

Figure 6: **Dual-band topological edge states in a single finite system: analysis in the linear regime.** In the considered lumped-element model, we require that the resonance frequency of the resonators $LF_n$ is lower than that of $HF_n$, i.e., $f_{LF} < f_{HF}$. Therefore, the resonators $HF_n$ exhibit mainly capacitance characteristics in the vicinity of $f_{LF}$, leading to the manifestation of only the resonance of $LF_n$ in the unit cell. Similarly, when close to the frequency $f_{HF}$ which is far from $f_{LF}$, the resonators $LF_n$ have solely mass behaviors. Collectively, our system is equivalent to a classic topological lattice made of single-resonant unit cells at two different frequencies, denoted as $f_L$ and $f_H$, which depends on the resonance frequencies $f_{LF}$ and $f_{HF}$, respectively, as delineated herein and detailed explained in Section B.1 above. Their mathematical derivations are provided in Appendix A.1.

Fig. 8: The influence of losses on the two topological edge states, where losses in the capacitors $C_a$ and in the resonators $LF_n$ are defined with respect to $Z_c$, the specific acoustic impedance of the air. On the contrary, since the resonators $HF_n$ are eventually implemented with actively controlled loudspeakers, their losses are quantified through a constant factor $\mu_R$ acting on the natural resistance of the loudspeakers. The results in Fig. 8 show that the edge state at a lower frequency $f_L$ is more sensitive to the losses in the system, especially in the dominant resonators $LF_n$. Conversely, the performance of the edge state at a higher frequency $f_H$ is more robust. It is relatively little affected by all potential losses, and we also performed active control to further minimize losses in the dependent resonators $HF_n$.

Fig. 9: Detailed theoretical results obtained by solving Eq. (A.2) with the Harmonic Balance Method (HBM). More cases are shown here compared to Fig. 2 in the main text.

Fig. 10: Theoretical results obtained by solving Eq. (A.2) with the time integration method (fourth-order Runge-Kutta). The evolutionary trends of the nonlinear edge state are consistent with those obtained with HBM (Fig. 9), except that the limit cases cannot be reached on

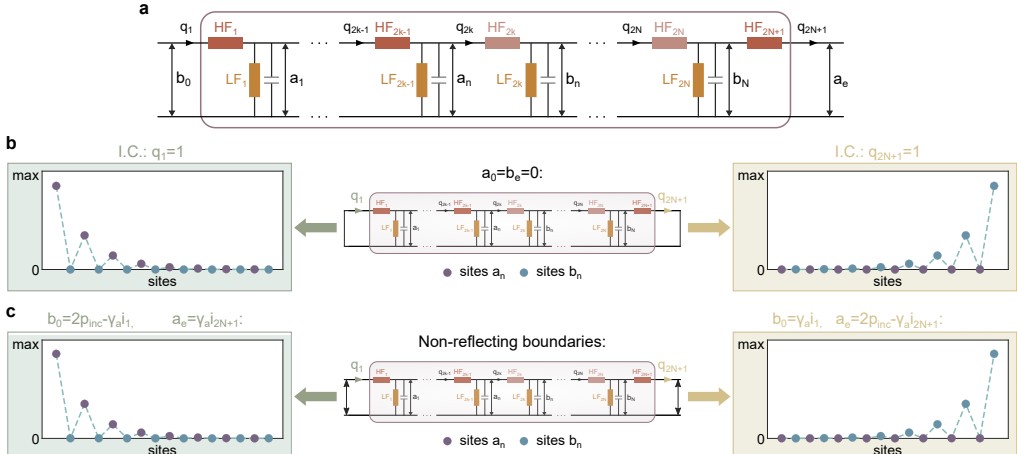

Figure 7: **Identification of realizable boundary conditions.** (**a**) The lumped element circuit considered, with $b_0$ and $a_e$ the input and output boundaries, respectively. $q_n$ designates the charge of the resonator $HF_n$. (**b**) Zero-energy topological edge state at $f_H$ derived with the ideal closed-closed boundary conditions ($b_0 = a_e = 0$), and with a nonzero initial conditions of $q_1 \neq 0$ (left inset) or $q_{2N+1} \neq 0$ (right inset), respectively. (**c**) Zero-energy topological edge state at $f_H$ derived with the Non-Reflecting Boundary Conditions (NRBCs) for both ends of the system (see Appendix A.2 for details), in which $i_{2N+1}$ ($i_1$) represents the electrical current circulating in $HF_{2N+1}$ ($HF_1$), and $\gamma_a = Z_c/S$ with $Z_c$ the specific acoustic impedance of the air and $S$ the surface area of the propagation medium. The excitation is defined at each of the two ends, respectively, through an incoming pressure source $p_{inc}$. All results are obtained with the 4-th order Runge-Kutta. They evidence the equivalence between the two types of boundary conditions. In this study, we opt for the NRBCs in (**c**) which is more realizable in our acoustic experiments.

account of the transition process. We demonstrate with simulations that reaching the plateau limit is actually possible (Appendix B.2, Fig. 12), as the hopping ratios are caused to be larger in the practical realizations.

Figs. 11: To better demonstrate the necessity of preserving chiral symmetry, we perform studies for cases where chiral symmetry is broken. Two nonlinearities are taken as examples:

(i), the first one is defined by

$$
\begin{cases}
V_{2k-1}^{(NL)} = +G_{NL}C^{(HF)}\beta_{2k-1}(b_{n-1}+a_n)^2(b_{n-1}-a_n)\,, \\
V_{2k}^{(NL)} = +G_{NL}C^{(HF)}\beta_{2k}(a_n+b_n)^2(a_n-b_n)\,,
\end{cases}
\tag{B.1}
$$

where the signs for $V_{2k-1}^{(NL)}$ and $V_{2k}^{(NL)}$ are all positive, as opposite to the nonlinearity with chiral symmetry where their signs are opposites.

(ii), the second one is

$$
\begin{cases}
V_{2k-1}^{(NL)} = -G_{NL}(b_{n-1}+a_n)^2(b_{n-1}-a_n)\,, \\
V_{2k}^{(NL)} = +G_{NL}(a_n+b_n)^2(a_n-b_n)\,,
\end{cases}
\tag{B.2}
$$

where the signs keep the same as the case with chiral symmetry, but the applied constant factors in $V_{2k-1}^{(NL)}$ and $V_{2k}^{(NL)}$ are modified.

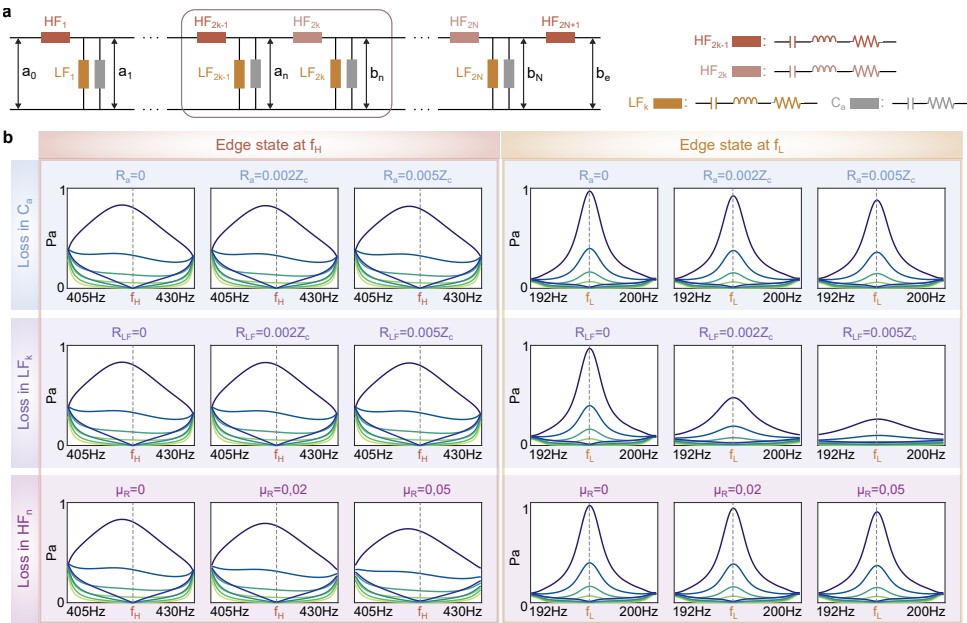

Figure 8: **Influence of losses on the two topological edge states at $f_H$ and $f_L$: the theoretical results when adding losses in each component.** Here we add losses to the resonators $HF_n$ (through $\mu_R$), the $LF_n$ (with $R_{LF}$) and the capacitors $C_a$ (with $R_a$), respectively, as sketched in (**a**). The spectra around the two corresponding linear edge states are given in (**b**).

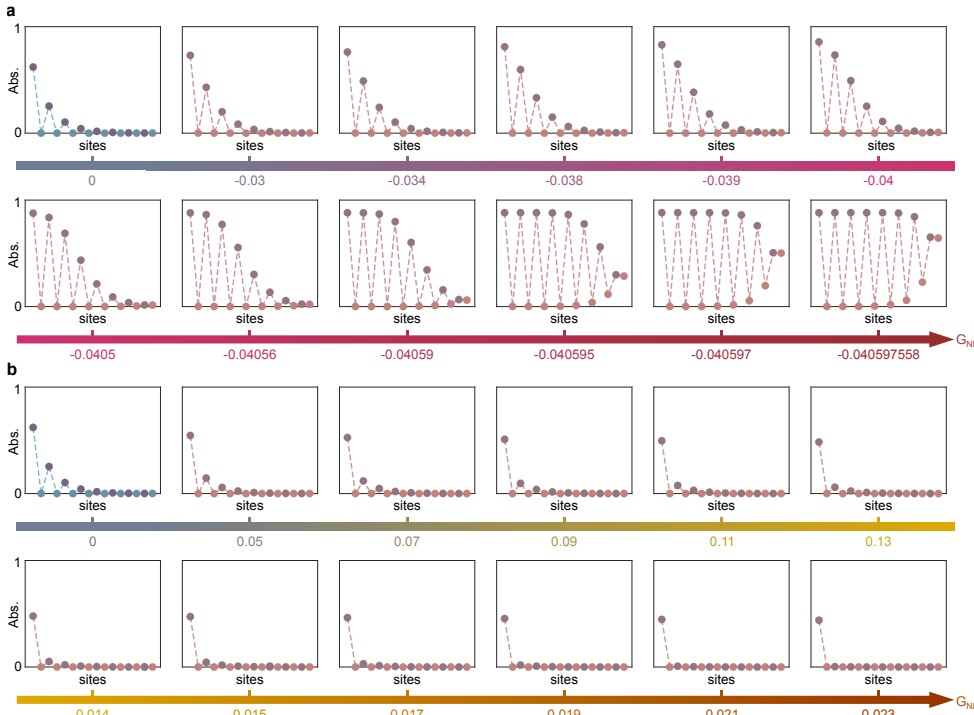

Figure 9: **Evolution of the chiral symmetry protected nonlinear topological edge states: detailed theoretical results.** The solutions are obtained with the Harmonic Balance Method (A.3). The level of nonlinearity is tuned using the constant parameter $G_{NL}$, the value of which varies in the negative (**a**) and positive (**b**) directions, respectively. All inset figures are displayed within the same amplitude range as in Fig.2 in the main text, while results of more nonlinear cases are showcased here.



Figure 10: **Evolution of the chiral symmetry protected nonlinear topological edge states: results obtained with the time integration method (fourth-order Runge-Kutta).** Cases of $G_{NL} < 0$ and $G_{NL} > 0$ are summarized in (**a**) and (**b**), respectively.

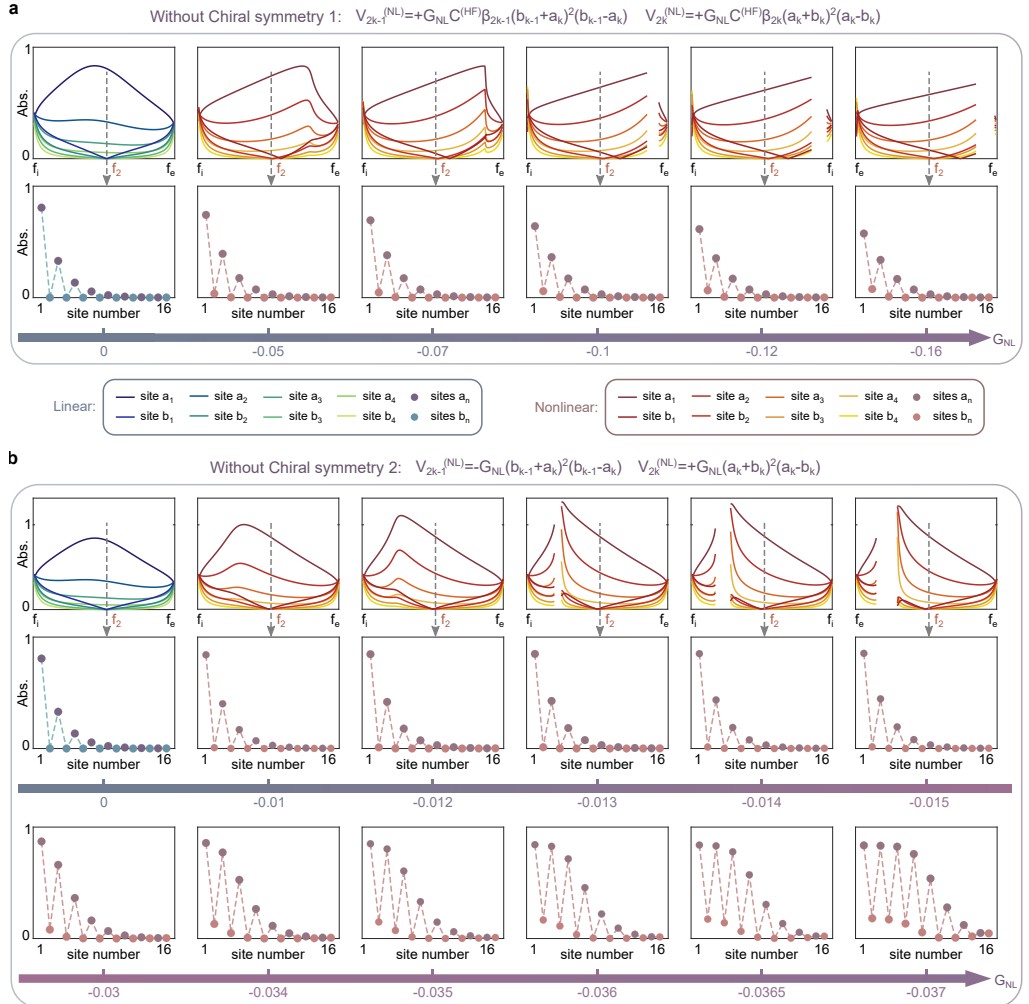

Figure 11: **Evolution of nonlinear topological edge state when nonlinearities break chiral symmetry: theoretical results.** Two forms of nonlinearities are investigated in (**a**) and (**b**), respectively. Results agree well with the numerical outcomes in Fig. 13 and the experimental ones in Fig. 16 where the same forms of nonlinearities are considered. They show that breaking chiral symmetry produces couplings between the two sublattices A and B, which causes the edge state to be shifted in frequency and distorted in shape.

The results in Fig. 11 demonstrate that, when nonlinearity breaks chiral symmetry, a coupling between the two sublattices A and B is created. Consequently, the sites $b_n$ in B move one by one together with those in A, which causes the edge state to be shifted in frequency and distorted in shape. The sane studies as in Fig. 11 are carried out numerically with time-domain simulations in Fig. 13 and experimentally with the acoustic system in Fig. 16, from which the same conclusion can be drawn.

## B.2 Simulation results

This section includes

Fig. 12: Detailed time-domain simulation results of the realized nonlinear topological edge states. Notably, practical situations are accounted for in the simulation (Appendix A.4), where

the pressure is not precisely homogeneous in each closed volume $V_a$, and the control uses the pressures close to each loudspeaker as inputs (Appendix A.6). In contrast to the simulations, the theoretical study assumes that the volume $V_a$ behaves as a capacitor $C_a$, thus presenting the same pressure over it. This eventually causes a difference in hopping ratios in the two studies. In theoretical results, the hopping ratio of the linear edge state is around 0.41, corresponding to the compliance ratio between $C_{2k}^{(HF)}$ and $C_{2k-1}^{(HF)}$. Whereas the linear hopping ratio obtained in simulations is equal to 0.52, larger than the theoretical value. The experimental value is 0.54, consistent with simulation outcomes.

Fig. 13: Simulation results for two cases where nonlinearities break chiral symmetry. They are in comparison with the theoretical ones in Fig. 11 and the experimental ones in Fig. 16, where the same forms of nonlinearities are considered. All studies show good agreement. They evidence that breaking chiral symmetry produces couplings between the two sublattices A and B, which causes the edge state to be shifted in frequency and distorted in shape.

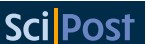

Figure 12: **Time-domain simulation results.** Results derived from time-domain simulation of the actual acoustic system. Nonlinearity adheres to chiral symmetry. The hopping ratios are (**a**) increased along $G_{NL} < 0$, and decreased along $G_{NL} > 0$, respectively.

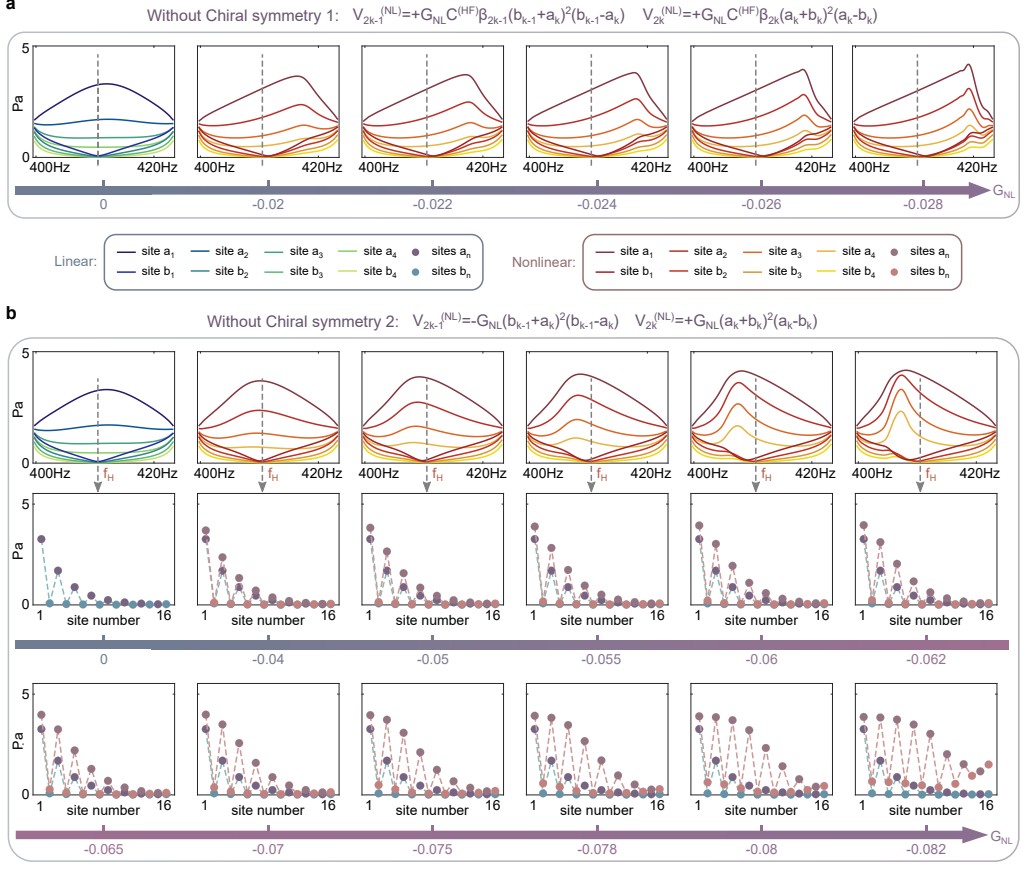

Figure 13: **Evolution of nonlinear topological edge state when nonlinearities break chiral symmetry: simulation results.** The actual acoustic system is simulated in the time domain. Two forms of nonlinearities are investigated in (**a**) and (**b**), respectively. Results agree well with the theoretical outcomes in Fig. 11 and the experimental ones in Fig. 16 where the same forms of nonlinearities are considered. They show that breaking chiral symmetry produces couplings between the two sublattices A and B, which causes the edge state to be shifted in frequency and distorted in shape.

## B.3 Experimental results

This section includes experimental results for the topological edge state at $f_H$ in Figs. 14, 15, 16, and for that at $f_L$ in Fig. 17 and 18, i.e.,

Fig. 14: Experimental results of the topological edge state at $f_H$ in the linear regime. In the frequency range of interest bounded by the dark dashed lines, the active (linear) control on the loudspeakers using Eq. (A.9) can achieve perfectly the impedance behaviors required by $HF_n$ in the theoretical model in Fig. 2a, despite the discrepancies in their natural (control-off) behaviors. The absorption coefficients of all loudspeakers are actively decreased from around 0.7 to less than 0.1 in the vicinity of $f_H$. The relevant topological edge state shows robustness also in responses to other actual losses (as evidenced in Fig. 8), thus it is eventually generated with barely any distortions in experiments.

Fig. 15: Detailed experimental results of the nonlinear topological edge state at $f_H$. More non-linear cases are shown compared to Fig. 3. They are in perfect agreement with both theoretical (in Fig. 9) and numerical (in Fig. 12) studies.

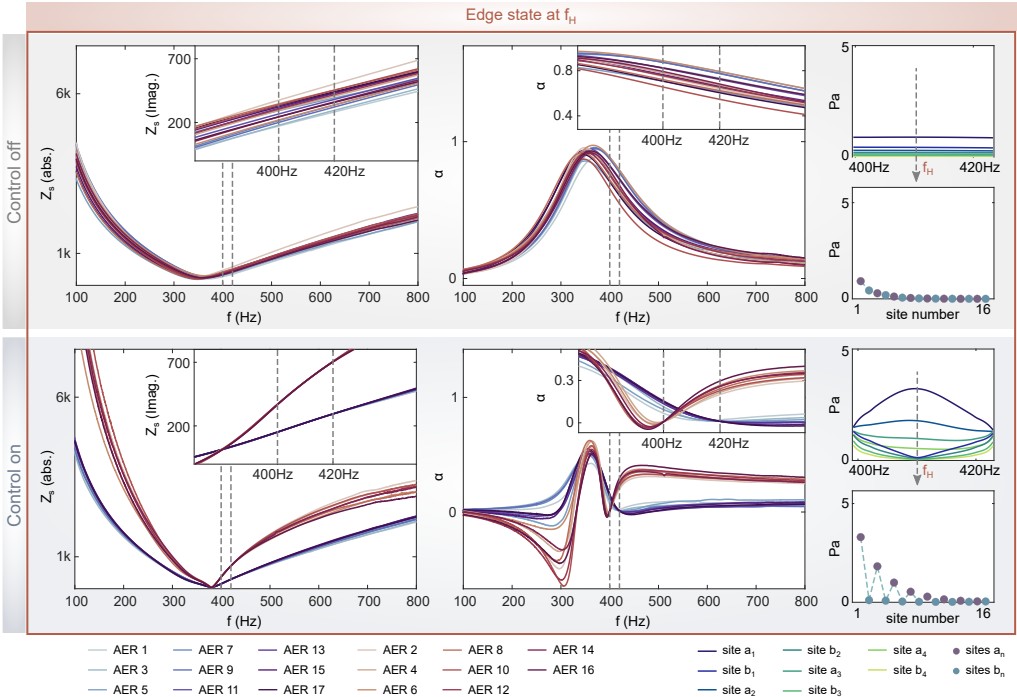

Figure 14: **Linear control results of topological edge state at $f_H$.** Comparison between the cases of control off and control on. The measured specific acoustic impedance $Z_S$ and absorption coefficient $\alpha$ are also illustrated in both cases, for all the 17 loudspeakers in use. The edge state is linearly generated without distortions.



Figure 15: **Evolution of the chiral symmetry protected nonlinear topological edge states: detailed experimental results.** More results are given here compared to Fig. 3 in the main text, for (**a**) $G_{NL} < 0$ and (**b**) $G_{NL} > 0$, respectively.

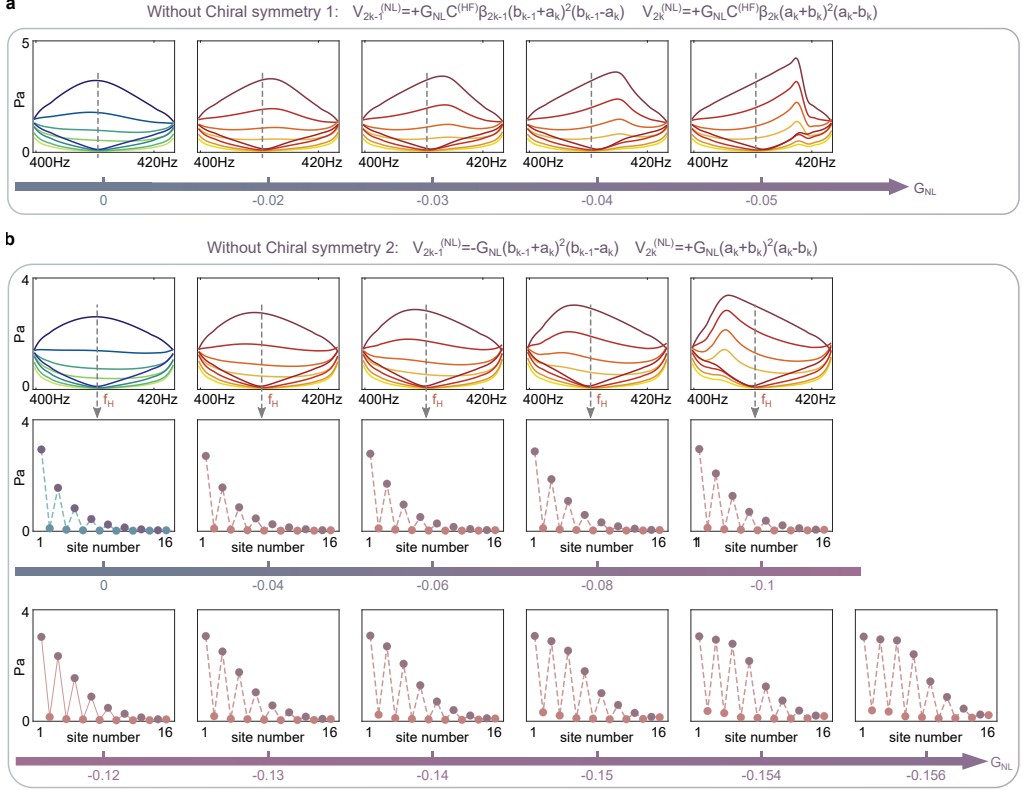

Figure 16: **Evolution of nonlinear topological edge state when nonlinearities break chiral symmetry: experimental results.** Two forms of nonlinearities are investigated in (**A**) and (**B**), respectively. Results agree well with the theoretical outcomes in Fig. 11 and the numerical ones in Fig. 13 where the same forms of nonlinearities are considered. They show that breaking chiral symmetry produces couplings between the two sublattices A and B, which causes the edge state to be shifted in frequency and distorted in shape.

Fig. 16: Experimental results for cases where nonlinearities break chiral symmetry. They are in comparison with the theoretical ones in Fig. 11 and the simulation ones in Fig. 13, where the same forms of nonlinearities are considered. All studies show good agreement. They evidence that breaking chiral symmetry produces couplings between the two sublattices A and B, which causes the edge state to be shifted in frequency and distorted in shape.

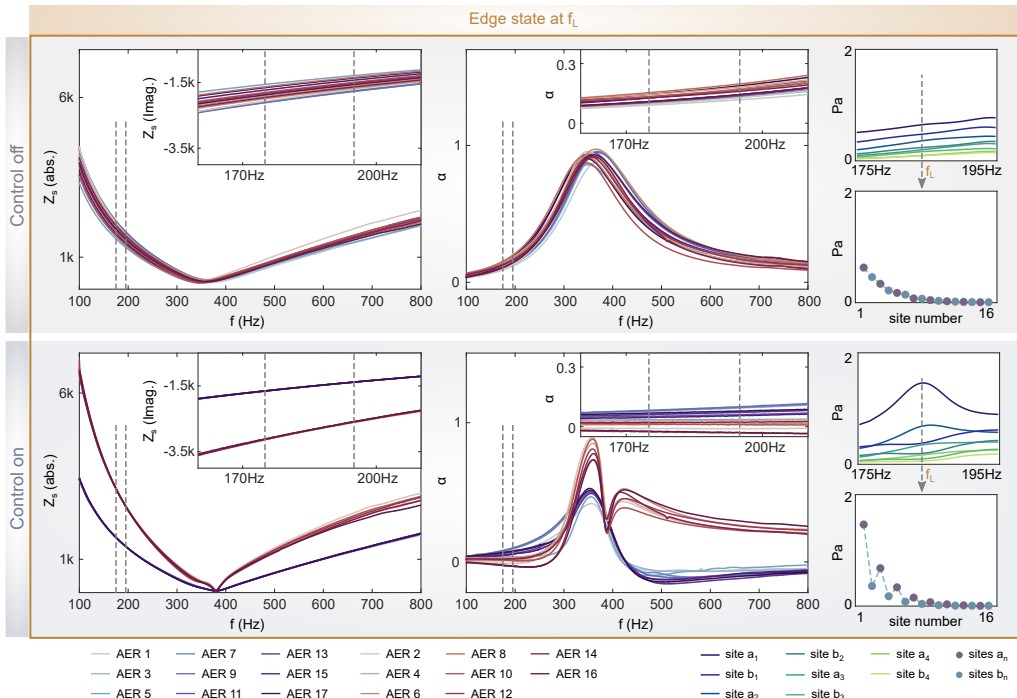

Figure 17: **Linear control results of topological edge state at $f_L$.** Comparison between the cases of control off and control on. The measured specific acoustic impedance $Z_S$ and absorption coefficient $\alpha$ are also illustrated in both cases, for all the 17 loudspeakers in use. The edge state is linearly generated with considerable distortions.

Fig. 17: Experimental results of the topological edge state at $f_L$ in the linear regime. The (linear) impedance properties (bandwidths and frequency) of the loudspeakers are tailored targeting a frequency range different from that of the previous edge state at $f_H$. The resonance of $LF_n$ plays a dominant role in this case, as explained in Fig. 6 in Appendix B.1. The associated edge state is sensitive to the losses in $LF_n$, as theoretically demonstrated in Fig. 8. Since $LF_n$ are realized with passive Helmholtz resonators in experiments, where losses unavoidably exist (even already small, see A.5), the edge state is unfortunately generated with noticeable distortion at $f_L$, i.e., the sites $a_n$ and $b_n$ are strongly coupled.

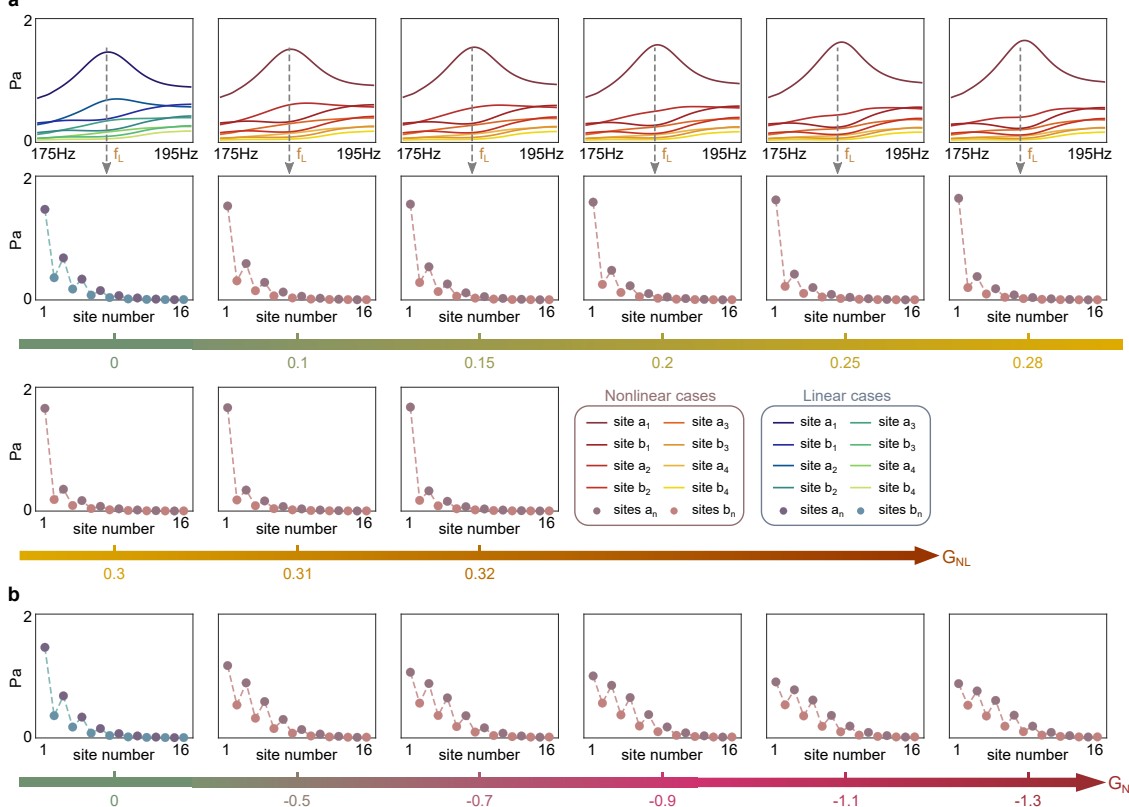

Figure 18: **Experimental results of the nonlinear topological edge states at** $f_L$**.** Results are shown for (**a**) $G_{NL} > 0$ and (**b**) $G_{NL} < 0$, respectively. Remarkably, the edge state at $f_L$ is severely distorted due to the actual losses in the system, even though the losses are already small.

Fig. 18: Detailed experimental results of the topological edge state at $f_L$ when nonlinearity is introduced. The nonlinear evolution of the edge state can be roughly identified along the direction of $G_{NL} > 0$, as can be noticed from the results in Fig. 18a. However, for $G_{NL} < 0$ (Fig. 18b), the edge state is dramatically destroyed, at which we fail to discover the expected nonlinear variation laws. This is mainly due to the loss effect already important in the linear regime (see Fig. 17), which remains consistently much stronger than the nonlinear effect after nonlinearity comes into play. Therefore, our study in the main part focuses on the other edge state at $f_H$, where the losses in the dominant elements (the resonators $HF_n$) can ultimately become negligible by applying active controls, resulting in perfectly intact nonlinear topological edge state.

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
