# Peer review of "Practical realization of chiral nonlinearity for strong topological protection"

_SciPost Physics, doi:SciPost Phys. 18, 034 (2025)_

## Round 1 · Referee Report · Anonymous (Referee 1) · 2024-10-22

Strengths
1. Clear understanding and presentation of the results.
2. A combination of theory, numerical results, and experiments to demonstrate topological edge states in the non-linear regime.
3. General arguments for the conditions under which chiral symmetry is preserved in the nonlinear regime, leading to a general argument for nonlinear topological states.
4. The timeliness of the topic, and the significance of the advance to clarify the conditions under which topology can be generalized to the nonlinear regime.
Weaknesses
1. The general context for the importance of nonlinear topological states and the nonlinear could be strengthened.
2. The fact that the authors only consider the nonlinear generalizations of SSH chains and chiral 1D systems is very restrictive. It is not obvious how to generalize the authors' approach to other topological invariants and other symmetry classes. However, this work could inspire future work in this direction.
3. The technological and practical motivation for creating nonlinear topological states could be elaborated or perhaps speculated.
4. The specific experimental realization might be difficult to generalize to other contexts beyond a circuit.
Report
In summary, the authors investigate the preservation of topological characteristics in systems with nonlinearity. The authors present a general theory for what can happen to topological states at high amplitudes due to the nonlinearity, and a condition under which a chiral symmetry can be defined even in the nonlinear regime. The authors then present a microscopic model and an experimental realization in terms of a lumped element circuit with coupled resonators, which they realize acoustically.
Overall, this is a nice addition to the literature. This work could motivate further advances in the timely area of nonlinear topology. The generalization of the symmetry condition into the nonlinear regime could have especially broad implications.
Recommendation
Publish (easily meets expectations and criteria for this Journal; among top 50%)
Author: Xinxin Guo on 2024-11-20 [id 4974]
(in reply to Report 1 on 2024-10-22)We sincerely thank the reviewer for the appreciation and acknowledgement of our work. We added more details about the system under consideration. We will continue to explore nonlinear topology and its applications, as encouraged by the reviewer.
Author: Xinxin Guo on 2024-11-20 [id 4973]
(in reply to Report 2 on 2024-10-24)Dear reviewer,
Please find attached the “Response to Comments.pdf” with our detailed responses to your questions and comments.
Best regards.
Attachment:
Responses_to_reviews.pdf

---

## Round 1 · Referee Report · Anonymous (Referee 2) · 2024-10-24

Strengths
1. Very interesting topic
2. Excellent work on the theoretical and experimental settings
Weaknesses
1. Lack of clarity at several points
Report
This is a very interesting and important work for the emerging field of nonlinear topological systems. I strongly recommend its publication but after some important revisions that will improve the clarity at several points
Requested changes
1. The first part is intended to be abstract and general. This approach is appropriate when the hopping ratios decrease with nonlinearity. However, in the case of increasing hopping ratios, a careful reading of the section reveals several additional conditions that have been established. For instance, there is a kind of saturable nonlinearity, where the nonlinearity must maintain the first hopping ratio at 1 once a2 =a1 is reached. Furthermore, they also consider the nonlinearity from the subsequent section. I recommend that the authors clarify that this is not a general case, but rather a specific scenario motivated by their physical context.
2. The authors aim to propose a physical system that exhibits the key features of chiral nonlinearity. However, the proposed setting is unnecessarily complicated. This is evident in Fig. 4, where at the frequency fH , the linear identical resonators (Helmholtz resonators) behave like an open circuit. By the way for the not specialists in electronics, the authors should clarify why this behavior occurs at high frequencies rather than at low frequencies. Wouldn't a variation of the approach presented in Phys. REVIEW APPLIED 20, 014022 (2023), using only membranes and active control, suffice to achieve the desired outcomes?
3. Considering a setting like finite SSH chains with medium hopping ratios, one would expect to observe two edge modes, coupled and with energies both below and above zero energy. It appears that the authors do not take this into account. Although the spectral figures, such as Fig. 2 and Fig. 3, show a dominant peak that is not at the frequency fH (the "zero" energy) but rather below it. The figure captions indicate that “the whole system starts and ends with the controlled loudspeakers,” suggesting that the authors are addressing a driven case rather than a Hamiltonian case, given the presence of loudspeakers at both ends of the structure. In this context, the edge profiles reflects the driven response of the system rather than the eigenmodes derived from the Hamiltonian's eigenvalues. This distinction may confuse the readers, especially since the first “general” section is based on the Hamiltonian framework. I recommend that the authors clarify these points in the text,
4. Lines 217-223 lack clarity, particularly regarding the authors' discussion of instability. While they reference several points related to instabilities, there are no results demonstrating how these instabilities are involved. For example, they state, “The limit of only a1 being dynamic cannot be observed, as instability arises first, which is in accordance with time-domain analysis (Appendix B.2, Fig. 9).” However, I do not see any evidence of instability dynamics in this figure. Identifying nonlinear unstable solutions is crucial, particularly in terms of how these instabilities manifest. Are the authors referring to wave instabilities or those arising from the active elements? Clarifying these points would enhance the reader's understanding.
5. In line 302, the authors state, “If Δt=0 is possible.” It is crucial for the validity of the results that Δt be zero, yet it is not clear under what conditions this occurs or if it is always the case. Please clarify this point clearly to ensure a better understanding of its implications for the results.
6. A final and significant comment: From Eq. A5, it appears that the nonlinearity results in non-reciprocal hopping. If this is indeed the case, the authors should address this explicitly. Please explain the constraints that led to this outcome. Additionally, could a reciprocal nonlinear chiral system be proposed along the lines of the electronic setup described in Nat Electronics 178, 2018, “Self-induced topological protection”? Addressing these points would enhance the comprehensiveness of the discussion
Recommendation
Ask for minor revision

---

## Round 2 · Referee Report · Anonymous (Referee 2) · 2024-11-29

Report

The authors have addressed all my points in a satisfactory manner, making the necessary changes where required. As a result, I believe the manuscript has been significantly improved and is now suitable for publication.

I therefore recommend the paper for publication in its current form.

Recommendation

Publish (easily meets expectations and criteria for this Journal; among top 50%)

---

## Editorial Decision

published